# ANCA: artificial nucleic acid circuit with argonaute protein for one-step isothermal detection of antibiotic-resistant bacteria

Hyowon Jang[1], Jayeon Song[1,2,3], Sunjoo Kim[4], Jung-Hyun Byun[4], Kyoung G. Lee [5], Kwang-Hyun Park [6], Euijeon Woo [6,7], Eun-Kyung Lim [1,8,9], Juyeon Jung [1,9] & Taejoon Kang [1,9] ✉

Endonucleases have recently widely used in molecular diagnostics. Here, we report a strategy to exploit the properties of Argonaute (Ago) proteins for molecular diagnostics by introducing an artificial nucleic acid circuit with Ago protein (ANCA) method. The ANCA is designed to perform a continuous autocatalytic reaction through cross-catalytic cleavage of the Ago protein, enabling one-step, amplification-free, and isothermal DNA detection. Using the ANCA method, carbapenemase-producing *Klebsiella pneumoniae* (CPKP) are successfully detected without DNA extraction and amplification steps. In addition, we demonstrate the detection of carbapenem-resistant bacteria in human urine and blood samples using the method. We also demonstrate the direct identification of CPKP swabbed from surfaces using the ANCA method in conjunction with a three-dimensional nanopillar structure. Finally, the ANCA method is applied to detect CPKP in rectal swab specimens from infected patients, achieving sensitivity and specificity of 100% and 100%, respectively. The developed method can contribute to simple, rapid and accurate diagnosis of CPKP, which can help prevent nosocomial infections.

Carbapenemase-producing Enterobacteriaceae (CPE) infections have emerged as a significant public health problem worldwide due to their rapid spread, high mortality and limited treatment options[1]. CPE is a group of Gram-negative bacteria, including *Klebsiella pneumoniae* carbapenemase (KPC), imipenemase (IMP), New Delhi metallo-beta-lactamase (NDM), Verona integron-encoded metallo-beta-lactamase (VIM), and oxacillinase-48 (OXA-48) producers, that exhibit hydrolytic activity against carbapenem antibiotics, the last line of defense against

multidrug-resistant bacterial infections[2]. The increasing prevalence of CPE infections in healthcare and community settings highlights the urgent need for rapid, accurate and cost-effective diagnostic methods to identify these resistant strains.

Current CPE detection methods, such as culture-based approaches and polymerase chain reaction (PCR)[3], are limited by turnaround time, labor intensity, and the need for specialized equipment and skilled personnel. Isothermal nucleic acid amplification methods offer

[1]Bionanotechnology Research Center, Korea Research Institute of Bioscience and Biotechnology (KRIBB), 125 Gwahak-ro, Yuseong-gu, Daejeon 34141, Republic of Korea. [2]Center for Systems Biology, Massachusetts General Hospital Research Institute, 175 Cambridge Street, Boston, MA 02114, USA. [3]Department of Radiology, Massachusetts General Hospital, Harvard Medical School, 55 Fruit Street, Boston, MA 02114, USA. [4]Department of Laboratory Medicine, Gyeongsang National University Hospital, Gyeongsang National University College of Medicine, 79 Gangnam-ro, Jinju-si, Gyeongsangnam-do 52727, Republic of Korea. [5]Division of Nano-Bio Sensors/Chips Development, National NanoFab Center (NNFC), 291 Daehak-ro, Yuseong-gu, Daejeon 34141, Republic of Korea. [6]Disease Target Structure Research Center, KRIBB, 125 Gwahak-ro, Yuseong-gu, Daejeon 34141, Republic of Korea. [7]Department of Biomolecular Science, KRIBB School of Biotechnology, University of Science and Technology (UST), 217 Gajeong-ro, Yuseong-gu, Daejeon 34113, Republic of Korea. [8]Department of Nanobiotechnology, KRIBB School of Biotechnology, UST, 217 Gajeong-ro, Yuseong-gu, Daejeon 34113, Republic of Korea. [9]School of Pharmacy, Sungkyunkwan University (SKKU), 2066 Seobu-ro, Jangan-gu, Suwon-si, Gyeongi-do 16419, Republic of Korea. ✉e-mail: kangtaejoon@gmail.com

an alternative diagnostic technique that can overcome these challenges and enable timely detection and appropriate treatment of CPE infections[4,5]. However, the complex composition of isothermal amplification mixtures often leads to incompatibility with clinical samples[6,7]. This requires additional purification procedures, equipment and reagents, ultimately limiting the practical applicability of isothermal nucleic acid diagnostics.

Recently, endonucleases with sequence-specific catalytic properties have emerged as valuable tools for nucleic acid target identification[8]. In particular, the clustered regularly interspaced short palindromic repeats (CRISPR)/CRISPR-associated protein (Cas) systems have attracted considerable attention for their potential in molecular diagnostics[9,10]. The CRISPR/Cas12 and CRISPR/Cas13 systems have demonstrated exceptional diagnostic potential due to their collateral cleavage activities[11]. Upon recognition and binding of target sequences, Cas12 and Cas13 exhibit non-specific cleavage of nearby single-stranded DNA and RNA molecules, respectively[12,13]. This collateral cleavage activity can be used for signal amplification, enabling highly sensitive detection of target nucleic acids[12–14]. Despite its promise, the diagnostic application of CRISPR/Cas systems faces several challenges, including the stringent requirement that the preferred cleavage site be in close proximity to the target sequence, as well as the increased cost and instability associated with the use of guide RNA[15,16]. To address these limitations, researchers are actively investigating strategies such as the development of alternative endonucleases and the use of chemically modified guide RNAs[17,18].

Argonaute (Ago) proteins, named for their homology to the Ago family of proteins in eukaryotes, play a critical role in RNA interference (RNAi) and small RNA-guided gene regulatory pathways[19]. These proteins are found in a wide range of organisms[19,20], including eukaryotes, prokaryotes and archaea, underscoring their evolutionary conservation and functional importance[21]. Their stringent DNA-guided target recognition capabilities and cleavage activities have made Ago proteins remarkable in bioanalysis. One interesting feature is their ability to cleave target nucleic acids without requiring a specific sequence motif, unlike the protospacer adjacent motif (PAM) used by Cas proteins[22,23]. Instead, Ago proteins achieve precise cleavage by base-pairing with a segment of guide DNA between positions 10 and 11[24,25], offering adaptability with less restrictive nucleic acid selection. Thermophilic variants of these proteins, derived from heat-loving microorganisms[26], add another layer of attractiveness due to their ability to withstand high temperatures, enhancing the robustness of Ago-based diagnostic methods. In addition, the use of a DNA-based guide probe, which is more stable and less expensive than RNA, can further increase the effectiveness of the assay.

In current research, Ago proteins are mainly used for direct cleavage after nucleic acid amplification reactions such as PCR, loop-mediated isothermal amplification (LAMP), etc. For example, refs. 27–29 used PCR to amplify specific target DNAs, which were then cleaved by Ago proteins, producing an enhanced fluorescence signal. Wang et al. [30] used reverse transcription (RT)-PCR to convert severe acute respiratory syndrome coronavirus 2 (SARS-CoV-2) RNA into cDNA. This cDNA was then subjected to two cleavage reactions triggered by Ago proteins, resulting in a fluorescent signal. Ye et al. [31] used LAMP technology to generate cDNA from SARS-CoV-2 and influenza RNA, followed by cleavage reactions similar to the method of Wang et al. Li et al. [32] combined exonuclease I with Ago proteins, amplified food poisoning bacterial DNA by PCR, and then generated a fluorescent signal by a cleavage reaction. In contrast, ref. 33 used Ago proteins for initial nucleic acid cleavage prior to pre-amplification. The resulting RNA fragment was then subjected to an exponential amplification reaction with dsDNA identification via SYBR I dye. Wang et al. [34] combined the ligation chain reaction with Ago proteins. This reaction joined two short strands, directed cleavage of the reporter probe, and produced a fluorescent result. Although these methods showed

promising detection capabilities by utilizing the superior target recognition capabilities of Ago proteins, they require an increased number of enzymes and procedural steps, resulting in a relatively complex process. Meanwhile, refs. 35,36 devised methods using Ago proteins without nucleic acid amplification, however, relatively poor performances were recorded.

Herein, we present a DNA detection technology that combines an artificial nucleic acid circuit with the programmable sequence-guided cleavage activity of Ago proteins. The artificial nucleic acid circuit with Ago protein (ANCA) method utilizes a rationally designed DNA complex with self-reporting capabilities, which efficiently incorporates the target recognition and cleavage activity of the Ago system. As a result, the ANCA can establish a positive feedback loop with high specificity and exponential signal amplification. Notably, the ANCA method relies on a single protein, which simplifies manufacturing and reduces costs compared to conventional nucleic acid detection circuits. The reaction is configured to run in a single step and can be easily adapted to detect different targets by changing the nucleic acid recognition site. We successfully detect carbapenemase-producing *Klebsiella pneumoniae* (CPKP) using the ANCA method. In addition, carbapenemase-producing bacteria spiked into human samples such as urine and blood are detected without the need for nucleic acid amplification and extraction steps. Furthermore, we demonstrate the ability to directly identify bacteria swabbed from surfaces such as pig skin, desk, glove, scissors, knob, and tweezers by using the ANCA method in conjunction with a three-dimensional (3D) nanopillar array structure. Most importantly, the ANCA assay is applied to diagnose CPKP-infected patients using rectal swab specimens, with clinical sensitivity and specificity of 100% and 100%, respectively. We anticipate that this approach has significant potential for point-of-care bacterial testing, facilitating patient care in the early stages of bacterial infections and ultimately improving diagnostic accuracy and treatment outcomes in clinical settings.

## Results
### Design of ANCA method
The ANCA assay is designed to exploit two distinct cleavage activities of the Ago protein. First, the cleavage site of the Ago protein is located at the 10th and 11th positions from the 5' end of the guide DNA[37]. Second, the Ago protein-cleaved strand can be used as a guide DNA because a phosphate group is attached to the 5' end of the strand[38]. The ANCA method was developed to make full use of these two cleavage activities to establish a cycle. The overall procedure of the ANCA method shown in Fig. 1 and sequence-based illustrations of the ANCA method are shown in Supplementary Figs. 1–5. For the implementation of the ANCA method, four signal processors-guide DNA 1 (G1), guide DNA 2 (G2), reporter (R), and reporter complement (R*)-were designed to work synergistically. G1 and G2 were designed according to the guide DNA design guidelines from the New England Biolabs Inc. (NEB) website. R contains both a quencher and a fluorophore. The sequences of all signal processors were carefully matched to ensure that Trigger 1 (T1) and Trigger 2 (T2), which are generated during the reaction process, could also serve as guide DNAs. The sequences of the oligonucleotides used in this experiment are listed in Supplementary Table 1.

The ANCA reaction begins when the Ago proteins form complexes with G1 and G2, respectively, and recognize the target DNA. The Ago/G1 and Ago/G2 complexes hybridize to the complementary target DNA sequences and then cleave the target nucleic acid between the 10th and 11th bases from the 5' end of each guide DNA (red symbols in Fig. 1). As a result, a short DNA fragment (T1) is generated from the target DNA. Since T1 itself can be used as another guide DNA, the Ago/T1 complex forms and recognizes and cleaves R, resulting in the release of two short DNA strands (Output and T2). The output material produces a fluorescence signal (dashed red box in Fig. 1), and T2 binds to the Ago protein to form the Ago/T2 complex. The newly formed

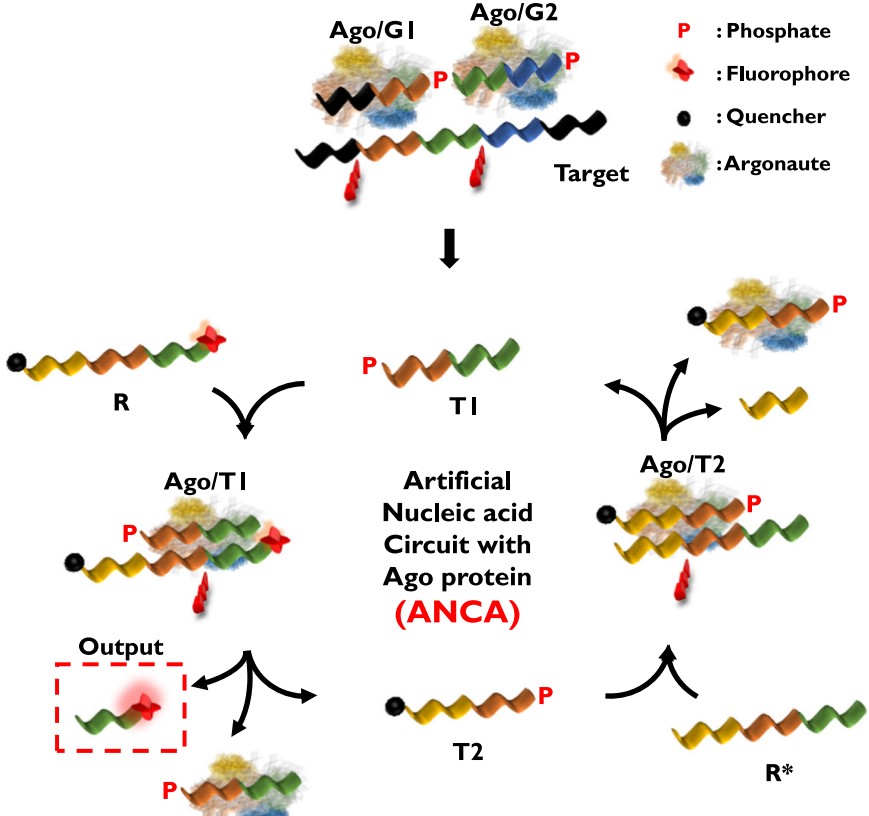

**Fig. 1 | Schematic illustration of ANCA method for the detection of target nucleic acid.** First, Ago/G1 and Ago/G2 complexes hybridize and cleave target DNA, generating T1. Second, Ago/T1 complex forms, and it hybridizes and cleaves R, releasing Output and T2. Third, Ago/T2 complex forms, and it hybridizes and cleaves R*, generating T1 and thus completing ANCA. In the presence of target DNA, ANCA is repeatedly carried out and fluorescence signal increases. P stands for phosphate attached to the 5′ end.

Ago/T2 complex recognizes and cleaves R*, generating T1 and completing the DNA cycle. Through this positive feedback loop, the cleavage reaction is repeated in the presence of target DNA, and as the reaction proceeds, the starting material is rapidly produced. Finally, the presence of target DNA in the sample can be determined by monitoring the fluorescence signal.

## Evaluation of ANCA method

The ANCA method was designed to provide exponential amplification through a positive feedback loop (Fig. 2a). We first evaluated the ANCA method with different components using KPC as the target. In the presence of all reaction components, including the target, strong fluorescence signals were observed during the ANCA reaction (spectrum 5 in Fig. 2c). In the absence of target DNA, Ago protein, G1, or G2, negligible signals were detected (spectra 1–4 in Fig. 2c). The feasibility of the ANCA method is further confirmed by the results of polyacrylamide gel electrophoresis (PAGE) analysis (right panel in Fig. 2c). The band indicating the intact R-R* structure (lane M2 in Fig. 2c) was completely degraded after the ANCA reaction (lane 5 in Fig. 2c). On the other hand, in the absence of target DNA, Ago protein, G1, or G2, the bands corresponding to the R-R* structure remain intact (lanes 1–4 in Fig. 2c). To evaluate the amplification efficiency of the ANCA method, we performed comparative experiments (Fig. 2b). In the ANCA reaction without R*, the Ago/G1 and Ago/G2 complexes cleave the target DNA, resulting in the formation of the Ago/T1 complex. The Ago/T1 complex then cleaves R, releasing Output and T2. However, no further reaction occurs because of the absence of R*. As a result, no fluorescence signals were observed even after 400 min of reaction, in the presence of the target sequence (spectrum 5 in Fig. 2d). In addition, no signals were detected in the absence of target DNA, Ago protein, G1, or

G2 (spectra 1–4 in Fig. 2d). The PAGE results also show the intact R structure bands for all conditions (right panel in Fig. 2d).

To verify that the higher concentration of target DNA could be detected with R alone, we repeated the experiment using a 10-fold increase in the concentration of target DNA used in Fig. 2. As a result, we could see that the fluorescence signal increased over time in the presence of R alone, although at a slower rate than in the presence of R and R* together (spectrum 5 in Supplementary Fig. 6). The PAGE results showed a fading of the band corresponding to R, where all components were present, and the production of a shorter band, which we speculate is due to the degradation of R (lane 5 in Supplementary Fig. 6). This evidence indicates that the ANCA method is a positive feedback circuit that provides high amplification efficiency and facilitates sensitive detection of target DNA.

## Optimization and sensitivity of ANCA method

To determine the optimal conditions for the ANCA method, we first monitored changes in fluorescence by varying R and R* (Supplementary Fig. 7). When the fluorophore and quencher were attached to the R sequence only, the ANCA method demonstrated efficient target DNA detection. It is worth noting that when the fluorophore and quencher were attached to both R and R*, the reaction efficiency was significantly reduced, which may be attributed to steric hindrance affecting the cleavage activity of Ago/guide DNA complexes. We then optimized the reaction temperature and the concentrations of R, R*, Ago protein, and $Mg^{2+}$ for the ANCA method. As shown in Supplementary Fig. 8a–e, the conditions for optimal detection performance were found to be a reaction temperature of 75 °C and concentrations of 500 nM for R and R*, 200 nM for Ago protein, and 10 mM for $MgCl_2$. In addition, the concentrations of NaCl and bovine serum albumin (BSA) in the enzyme

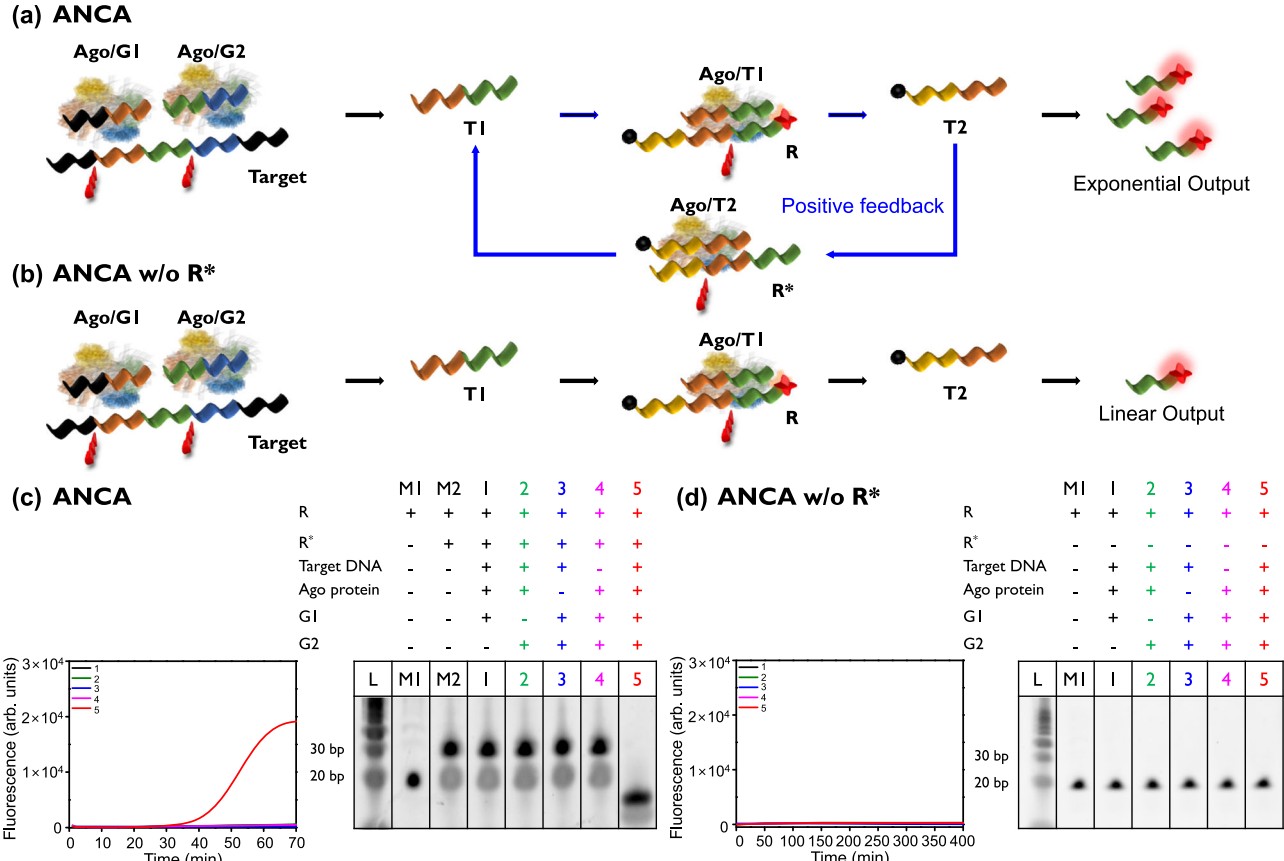

**Fig. 2 | Evaluation of ANCA method.** Schematic illustration of (**a**) ANCA and (**b**) ANCA without R*. ANCA method enables exponential amplification through positive feedback. ANCA without R* cleaves target DNA and R, but no further reaction transpires. **c** Time-dependent fluorescence intensities during ANCA reaction (left) and corresponding PAGE analysis result (right) under various components. **d** Time-dependent fluorescence intensities during ANCA reaction without R* (left) and corresponding PAGE analysis result (right) under various components. [Target DNA] = 1 nM, [R] = 500 nM, [R*] = 500 nM, [G1] = 25 nM, [G2] = 25 nM, [Ago] = 200 nM, [MgCl$_2$] = 10 mM, [NaCl] = 75 mM, and [BSA] = 10 μg/mL. The experiments were independently replicated three times.

storage buffer were optimized to 75 mM and 10 μg/mL, respectively (Supplementary Fig. 8f, g). These conditions were used in subsequent experiments.

Under the optimized conditions, we evaluated the sensitivity of the ANCA method for KPC and IMP as target sequences, respectively. Fig. 3a shows the time-dependent fluorescence intensities for different concentrations of the KPC sequence. The threshold time was determined as the reaction time at which the fluorescence intensity reached 10,000. Fig. 3b shows the plot of threshold time as a function of target DNA concentration. A linear relationship ($R^2 = 0.99$) was observed in the range of 10 fM to 10 nM, and the limit of detection (LOD) was calculated to be 1.87 fM using the formula[39] (limit of blank (LOB) = mean of blank + 1.645 × standard deviation of blank, LOD = LOB + 1.645 × standard deviation of low concentration sample). Additionally, we adapted the ANCA method for the detection of IMP. Using the IMP circuit, a linear relationship ($R^2 = 0.99$) was observed in the range of 1 fM to 1 nM, and the LOD was calculated to be 178 aM (Fig. 3c, d). Similarly, circuits for the detection of VIM, NDM, and OXA-48 were fabricated and evaluated for their sensitivity to each of the target substances. In all cases, a linear relationship ($R^2 = 0.99$) was observed in the range of 1 fM to 1 nM, and the LOD of VIM, NDM, and OXA-48 were calculated to be 529, 120, and 144 aM, respectively (Supplementary Fig. 9). The sensitivity of the ANCA method is superior to previous results that utilize Ago proteins without pre-amplification (Supplementary Table 2)[27–36]. Furthermore, it is noteworthy that the ANCA method demonstrates flexibility and broad target applicability, as evidenced by the successful detection of KPC, IMP, VIM, NDM, and OXA-48 sequences by slight modifications of the ANCA method.

## Direct detection of antibiotic-resistant bacteria using ANCA method

We attempted to detect CPKP by the ANCA method without DNA purification. KPC-producing *K. pneumoniae* (*K. pneu* (KPC)) and IMP-producing *K. pneumoniae* (*K. pneu* (IMP)) were cultured separately to avoid cross-contamination and used in the experiment. The cultured bacteria were verified by PCR using KPC- and IMP-specific primers (Supplementary Fig. 10). As shown in Fig. 4a, antibiotic-resistant bacteria were added directly to the tube containing the ANCA reaction mixture and incubated at 75 °C. Fig. 4b shows the bacterial detection results for the KPC circuit in the presence of *K. pneu* (KPC), *K. pneu* (IMP), and wild-type *K. pneumoniae* (*K. pneu* (WT)). T$_0$ represents the threshold time for the negative control sample (absence of bacteria), while T represents the threshold time for the test sample (presence of bacteria). The real-time fluorescence results are also shown in Supplementary Fig. 11a. The highest T$_0$–T value was observed only in the presence of *K. pneu* (KPC) (blue bar in Fig. 4b), confirming the direct detection of the target antibiotic-resistant bacteria using the ANCA reaction for KPC. Fig. 4c and Supplementary Fig. 11b show the detection results of IMP circuit in the presence of *K. pneu* (KPC), *K. pneu* (IMP), and *K. pneu* (WT), demonstrating the direct identification of *K. pneu* (IMP). We also attempted to detect CPKP using the VIM, NDM, and OXA-48 circuits and found that each circuit was able to discriminate well between WT and CPKP. (Supplementary Fig. 12a–c). The

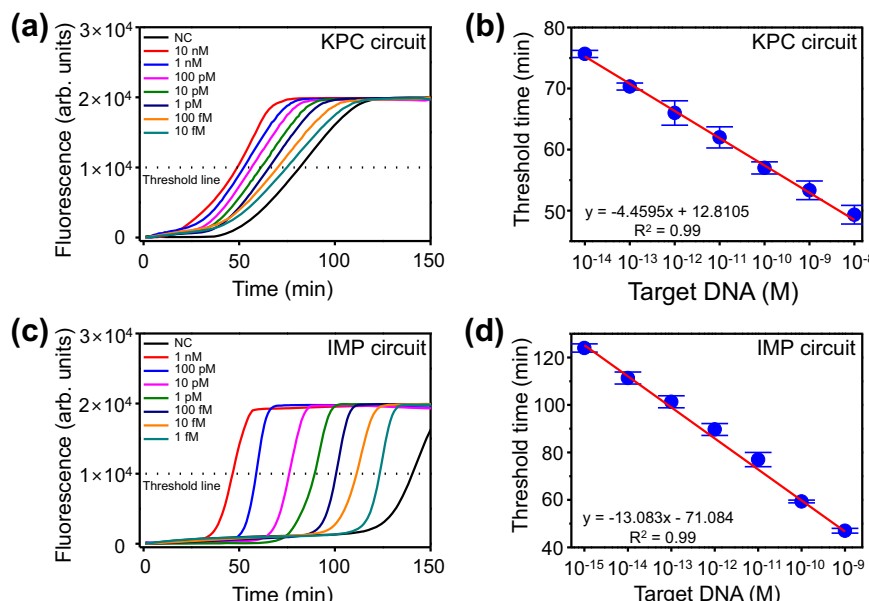

**Fig. 3 | Sensitivity of ANCA method.** Time-dependent fluorescence intensities during ANCA method with various concentrations of (**a**) KPC and (**c**) IMP sequences. Dashed black threshold lines indicate the reaction time at which the fluorescence intensity reached 10,000 (threshold time). Correlation of threshold time to the logarithm of (**b**) KPC and (**d**) IMP concentration ($n = 3$ independent experiments, error bar = standard deviation). Data are presented as mean values +/− standard deviation. Red lines are linear fits, indicating LODs of 1.87 fM for KPC and 178 aM for IMP. [R] = 500 nM, [R*] = 500 nM, [G1] = 25 nM, [G2] = 25 nM, [Ago] = 200 nM, [MgCl$_2$] = 10 mM, [NaCl] = 75 mM, and [BSA] = 10 µg/mL.

ANCA method allows for the detection of CPKP in an simplified manner, without the need for pre-concentration, lysis, and DNA purification steps. This can be attributed to the robustness of the ANCA method at a high reaction temperature (75 °C). According to previous literature[40], bacteria can be lysed within 15 min at temperatures above 60 °C. Therefore, it is possible to directly detect CPKP using the ANCA method. Considering that the lysis and nucleic acid extraction steps are complex, difficult, and time-consuming, the ANCA method, which eliminates these processes, is advantageous for simple and rapid diagnosis of infectious bacteria.

To investigate the ANCA method for the detection of bacteria in clinical specimens, we prepared CPKP-spiked human urine and blood samples. These samples were first tested by PCR using a typical DNA extraction kit (Supplementary Fig. 13). In addition, we performed tests on urine and blood samples to determine the robustness of the experiment against autofluorescence. When evaluating the fluorescence signal within the FAM wavelength range, we observed minimal differentiation between the samples and distilled water (Supplementary Fig. 14). Next, the bacteria-spiked urine and blood samples were added directly to the tube containing the ANCA reaction mixture and incubated (Fig. 4d–g). Remarkably, the KPC circuit was able to detect *K. pneu* (KPC) in 90% (Supplementary Fig. 15a) and even 99% of human urine (Fig. 4e). The IMP circuit also successfully detected *K. pneu* (IMP) in urine (Fig. 4f and Supplementary Fig. 15b). Furthermore, the ANCA method facilitated the identification of *K. pneu* (KPC) and *K. pneu* (IMP) in 90% and 99% of blood samples, respectively (Fig. 4h, i, Supplementary Fig. 15c, d). This suggests that the ANCA method can be used for the detection of CPKP under harsh conditions. Given that monitoring CPKP in urine and blood samples can provide valuable epidemiological data on the prevalence and distribution of antibiotic-resistant bacteria[41,42], the present results have important implications.

### Diagnosis of antibiotic-resistant bacteria using ANCA method

The standard diagnostic procedure for CPE involves several steps, including sample collection, bacterial culturing, colony identification and isolation, antibiotic susceptibility testing, and determination of carbapenem resistance[43,44]. The entire process takes several days,

highlighting the need for rapid and efficient methods to identify antibiotic-resistant bacteria. Typically, CPE is diagnosed using rectal swabs collected from patients[45]. To ensure the viability and stability of the collected rectal swab sample, the swab is immediately immersed in transport media after collection. Transport media are specifically formulated to maintain the integrity of the bacterial sample during transportation and storage, thus preserving the characteristics of the sample and preventing overgrowth of competing microorganisms[46]. We attempted to diagnose CPKP from clinical samples using the ANCA method. A total of 143 rectal swabs (63 positive for KPC and 80 negative) were collected from patients using cotton swabs, and each swab was placed in transport media. As shown in Fig. 5a, the rectal swab-immersed transport media was directly combined with the KPC circuit mixture and incubated under fluorescence monitoring. Remarkably, the ANCA method was able to differentiate 63 KPC-positive samples from 79 negative samples based on the cut-off line (Fig. 5b). Compared to the hospital diagnostic results, a false-positive signal was detected only from clinical sample #2. In this experiment, $T_0$ represents the threshold time for the pure transport media and $T$ represents the threshold time for the rectal swab-immersed transport media. The cut-off $T_0 − T$ value was calculated as 10.11 using the formula[47,48] (cut-off $T_0 − T$ value = mean $T_0 − T$ value of 80 negative samples +5 × mean standard deviation of 80 negative samples). The ANCA-based diagnostic tests were performed in a blinded manner and the positive samples were labeled from #81 to #143 based on the $T_0 − T$ value after the experiments. Additionally, we performed a comparative analysis between the diagnostic results of the ANCA method and those of PCR. A strong correlation was observed between $T_0 − T$ and $C_{t0} − C_t$ values for KPC-positive clinical samples (Fig. 5c). $C_{t0}$ represents the cycle threshold of the pure transport media and $C_t$ represents the cycle threshold of the clinical sample. This result indicates that the developed ANCA method can be used for simple and accurate diagnosis of CPKP.

To assess the clinical sensitivity and specificity of the ANCA method, we used 57 bacteria-spiked rectal swab samples in addition to the clinical samples. Since the clinical sensitivity and specificity of molecular diagnostics approved by the Korean Food & Drug

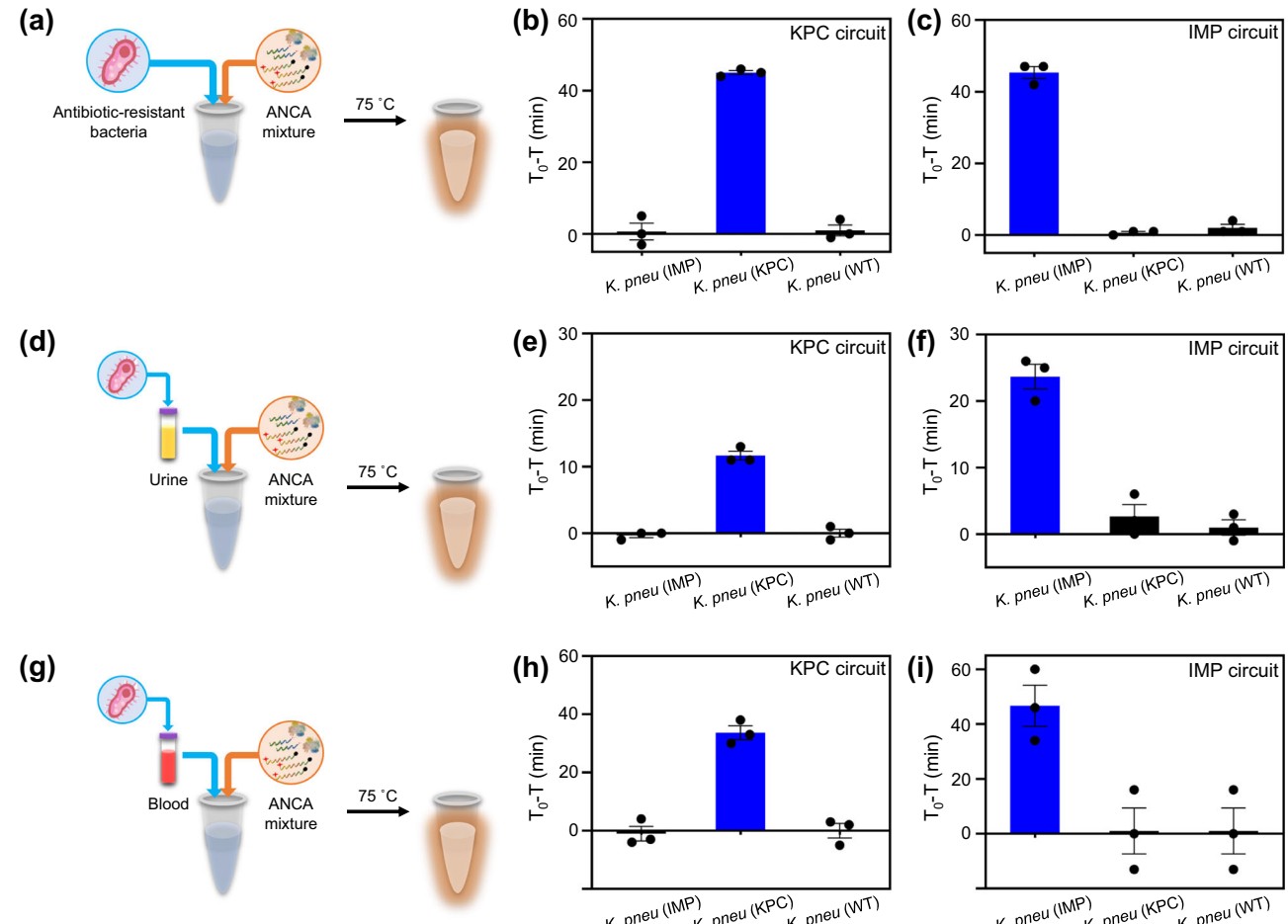

**Fig. 4 | Direct detection of antibiotic-resistant bacteria using ANCA method.** Schematic illustration for the direct detection of CPKP in (**a**) buffer, (**d**) urine (99%), and (**g**) blood (99%) using ANCA method. Plot of $T_0$ – T value as a function of bacteria in buffer using (**b**) KPC and (**c**) IMP circuits ($n = 3$, error bar = standard deviation). Plot of $T_0$ – T value as a function of bacteria in urine (99%) using (**e**) KPC and (**f**) IMP circuits ($n = 3$, error bar = standard deviation). Plot of $T_0$ – T value as a function of bacteria in blood (99%) using (**h**) KPC and (**i**) IMP circuits ($n = 3$ independent experiments, error bar = standard deviation). $T_0$ represents the threshold time for negative control sample (absence of bacteria), and T denotes the threshold time for test sample (presence of bacteria). Data are presented as mean values +/– standard deviation.

Administration (KFDA) are 95% each[49], it is necessary to test at least 80 positive samples using the ANCA method. Therefore, we prepared additional mock clinical samples by spiking *K. pneu* (KPC) into negatively diagnosed rectal swab samples. The spiked amounts of *K. pneu* (KPC) were randomly determined and the tests were conducted in a blinded manner. Supplementary Fig. 16a shows the diagnostic results of a total of 200 clinical and the bacteria-spiked samples, indicating a clinical sensitivity and specificity of 100% and 98.7%, respectively. The plot of $T_0$ – T value as a function of clinical and bacteria-spiked sample is shown in Supplementary Fig. 16b. The mock clinical samples were labeled from #144 to #200 based on the $T_0$ – T value after the experiments. A receiver operating characteristic (ROC) analysis yielded an area under the curve (AUC) value of 0.9997, indicating high accuracy of the ANCA method compared to PCR (Fig. 5d). These results are remarkable considering that the ANCA method uses clinical samples directly without DNA purification and amplification steps.

A total of 160 clinical and mock clinical specimens (80 positive for IMP and 80 negative) were used to evaluate the diagnostic performance of the IMP circuit. Although patients were followed in the hospital for several months to collect IMP-positive rectal swab samples, no infected patients were found. The IMP-positive samples were prepared by randomly adding *K. pneu* (IMP) to negatively diagnosed rectal swab-immersed transport media. The diagnostic results for *K. pneu* (IMP) show that the IMP circuit has a clinical sensitivity and

specificity of 100% and 100%, respectively, with an AUC of 1.0 (Fig. 5e, Supplementary Fig. 16a, c).

The ANCA method exhibits a robust ability to detect antibiotic-resistant bacteria under challenging conditions due to its exceptional stability. In response, we sought to directly detect *K. pneu* (KPC) from cotton swabs obtained from patients. The swab was placed directly into the tube containing the KPC circuit mixture and the reaction was run at 75 °C (Fig. 5f). Even when rectal swabs were used directly, the ANCA method was successful in identifying *K. pneu* (KPC) (Fig. 5g). It is noteworthy that diagnostic outcomes derived from rectal swabs (sensitivity = 100%, specificity = 100%) showed superior accuracy compared to those from rectal swab-immersed transport media (sensitivity = 98.7%, specificity = 100%). Moreover, we observed a strong correlation between the $T_0$ – T values of the KPC-positive rectal swabs and the transport media (Supplementary Fig. 17). This suggests that the collected CPKP was simultaneously present on the rectal swab and in the transport media. Interestingly, the average $T_0$ – T value of the KPC-positive rectal swabs (54.89) was higher than that of the transport media (47.02) (Fig. 5i). This indicates a higher presence of *K. pneu* (KPC) on the rectal swabs than in the rectal swab-immersed transport media. To quantitatively compare the amounts of target DNA between sample types, we used the calibration curve of the KPC circuit shown in Fig. 3b. For ease of calculation, we postulated that the $T_0$ –

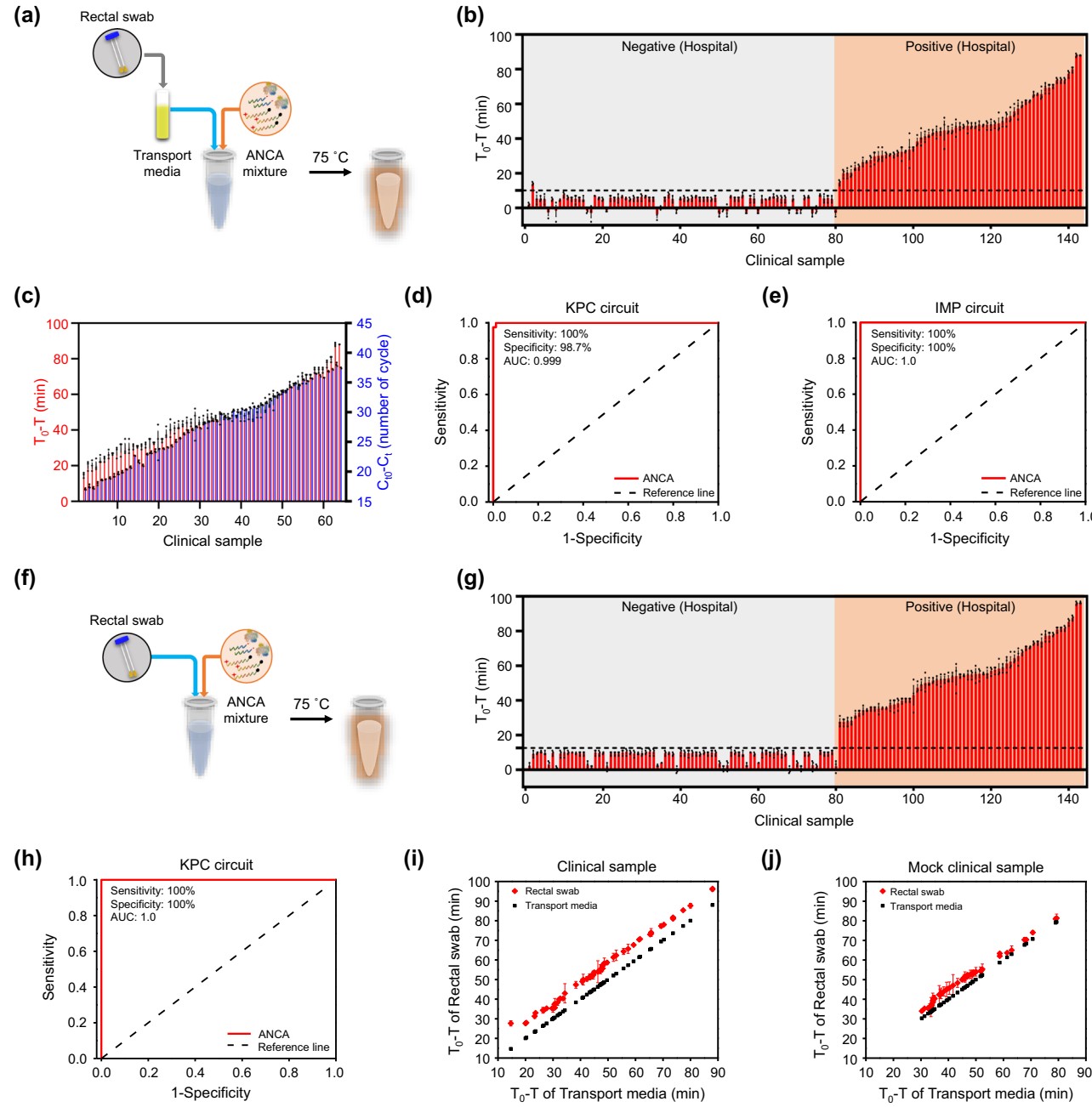

**Fig. 5 | Diagnosis of antibiotic-resistant bacteria using ANCA method.**
**a** Schematic illustration for the diagnosis of CPKP from rectal swab-immersed transport media using ANCA method. **b** Plot of $T_0 - T$ value as a function of clinical sample (rectal swab-immersed transport media) ($n = 3$ independent experiments, error bar = standard deviation). $T_0$ represents the threshold time for pure transport media, and T denotes the threshold time for clinical sample. Cut-off $T_0 - T$ value (10.11) is marked as dashed black line. Data are presented as mean values +/− standard deviation. **c** Comparative plot of $T_0 - T$ (red) and $C_{t0} - C_t$ (blue) values as a function of KPC-positive clinical sample ($n = 3$ independent experiments, error bar = standard deviation). $C_{t0}$ represents the cycle threshold for pure transport media and $C_t$ denotes the cycle threshold for clinical sample. Data are presented as mean values +/− standard deviation. **d** ROC curves generated from the diagnostic results of a total of 200 clinical (143) and *K. pneu* (KPC)-spiked (57) samples (rectal swab-immersed transport media). Clinical sensitivity and specificity are 100% and 98.7%, respectively. **e** ROC curves generated from the diagnostic results of a total of 160 clinical (80) and *K. pneu* (IMP)-spiked (80) samples (rectal swab-immersed transport media). Clinical sensitivity and specificity are 100% and 100%, respectively. **f** Schematic illustration for the diagnosis of CPKP from rectal swab directly using ANCA method. **g** Plot of $T_0 - T$ value as a function of clinical sample (rectal swab) ($n = 3$ independent experiments, error bar = standard deviation). Data are presented as mean values +/− standard deviation. $T_0$ represents the threshold time for pristine rectal swab, and T denotes the threshold time for clinical sample. Cut-off $T_0 - T$ value (12.55) are marked as dashed black line. **h** ROC curves generated from the diagnostic results of a total of 200 clinical (143) and *K. pneu* (KPC)-spiked (57) samples (rectal swab). Clinical sensitivity and specificity are 100% and 100%, respectively. Scatter plot of $T_0 - T$ values obtained from rectal swab (red) to the values from transport media (black) using KPC-positive (**i**) clinical and (**j**) mock clinical samples ($n = 3$ independent experiments, error bar = standard deviation). Data are presented as mean values +/− standard deviation.

T value would demonstrate a trend akin to the $y$ value of the calibration curve. This led us to the following equation: $y_t - y_s = -4.4595(x_t - x_s)$, where $y_t$ and $y_s$ represent the $T_0 - T$ values of the transport media and swab, respectively, and $x_t$ and $x_s$ represent the log of KPC concentration in the transport media and swab, respectively. By substituting $y_t = 47.02$ and $y_s = 54.89$ into the equation, we derived a value of $x_t - x_s = 1.764$. Given that $x$ signifies the value derived from the logarithm of target nucleic acid concentration, the actual concentration ratio between the two samples is 58.076. This implies that KPC-producing bacteria can be detected over 50 times more sensitively when a rectal swab is used directly. A similar trend was observed in the experimental results derived from mock clinical samples (Fig. 5j). In these comparative experiments, rectal swabs were initially immersed in transport media prior to being used in the ANCA method. Given that the volume of transport media (5 mL) is large enough to dilute the bacterial concentration, these results are consistent with logical expectations. Nevertheless, it is important to note that a significant number of bacteria remained on the rectal swabs even after immersion into transport media. These observations suggest that the ANCA method holds great promise for the direct detection of antibiotic-resistant bacteria from rectal swabs. This may help to achieve accurate diagnostic results and promote appropriate treatment strategies in healthcare settings.

## Capture and detection of antibiotic-resistant bacteria using ANCA method with 3D nanopillar swab

Nosocomial infections can be transmitted by a variety of routes, including direct and indirect contact with infected individuals, respiratory droplet or airborne transmission, and other vehicular transmission[50]. In particular, CPE transmission often occurs via contaminated skin and medical equipment[51]. Therefore, regular monitoring of patients and contaminated surfaces can aid in the early detection and control of potential CPE outbreaks. Since the ANCA method has been shown to detect KPC- and IMP-producing bacteria from clinical specimens, we further aimed to capture and detect antibiotic-resistant bacteria from contaminated surfaces using the ANCA method (Fig. 6a). To efficiently capture bacteria from surfaces, a 3D nanopillar array swab was employed. We first selected micropig skin as a substitute for human skin, which is reported to have properties very similar to human skin, to create an environment where direct contact occurs. We also selected a desk, glove, scissors, knob, and tweezers as experimental materials to create an environment where indirect contact occurs. These objects are mentioned in the Korean Standard Prevention Guidelines for Healthcare Associated Infections as requiring frequent disinfection. For the experiments, KPC- and IMP-producing bacteria were intentionally sprayed on micropig skin, desk, glove, scissors, knob, and tweezers. The swabs were then used to capture bacteria on these surfaces by gently touching and rubbing them. Subsequently, the bacteria-captured swab was placed directly into a tube and the ANCA reaction was carried out. The complex 3D nanopillar array structure facilitates bacterial adhesion through nanotopographical interactions, allowing irreversible capture of bacteria and thus preventing secondary infection[52]. Fig. 6b shows the scanning electron microscope (SEM) images of bare and CPKP-captured complex nanopillar array structures. Bacteria were successfully captured on the swab. Figure 6c, d show the results of KPC and IMP detection using the ANCA method for each target. For the KPC circuit, fluorescence signals were rapidly amplified only when *K. pneu* (KPC) was captured by the 3D nanopillar swab. Similarly, for the IMP

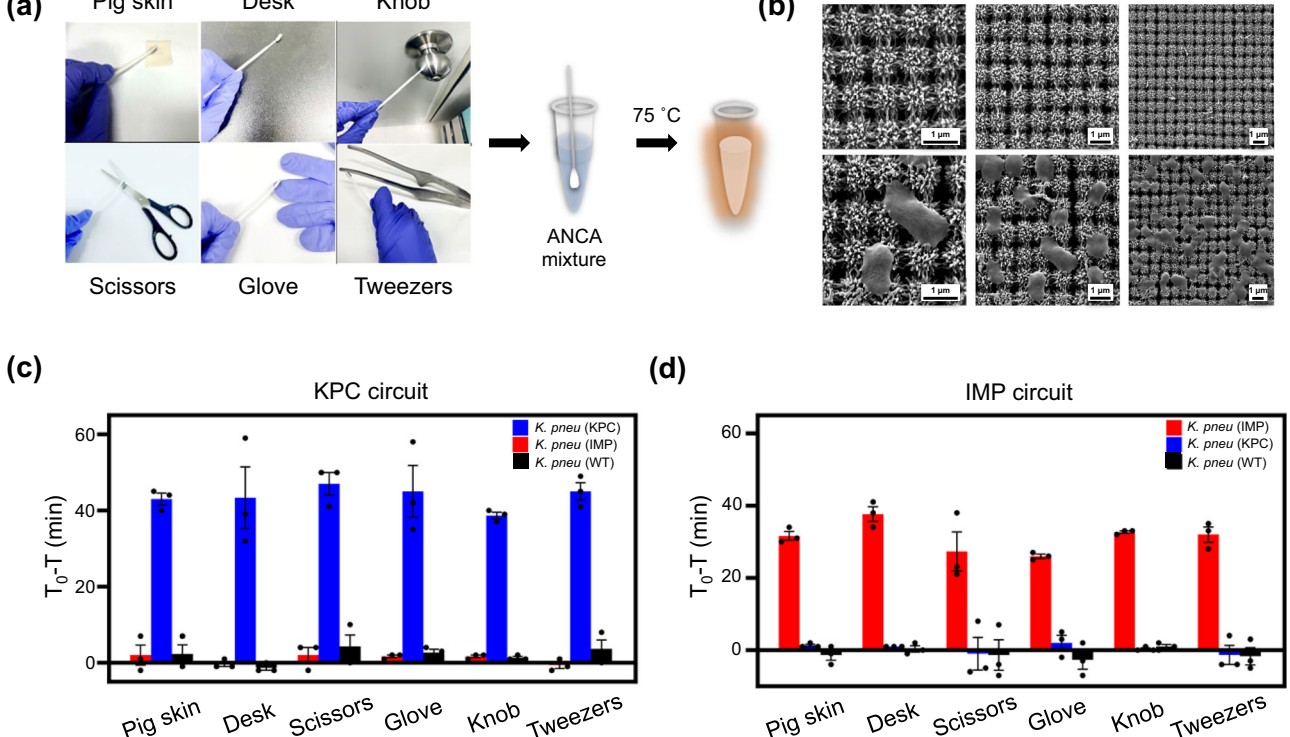

**Fig. 6 | Capture and detection of antibiotic-resistant bacteria using ANCA method with 3D nanopillar swab. a** Photograph of 3D nanopillar array swabs for the capture of CPKP from micropig skin, desk, scissors, glove, knob, and tweezers. Each bacterium was intentionally sprayed on the surfaces. CPKP-captured swab was directly used for ANCA reaction. **b** SEM images of 3D nanopillar array swabs before (upper) and after (lower) the capture of CPKP. The experiments were independently replicated three times. Plot of T0 − T value as a function of contaminated surface using (**c**) KPC and (**d**) IMP circuits (*n* = 3 independent experiments, error bar = standard deviation). T0 represents the threshold time for negative control sample (absence of bacteria), and T denotes the threshold time for test sample (presence of bacteria). Data are presented as mean values +/− standard deviation.

circuit, signals increased only when *K. pneu* (IMP) was captured. These results suggest that on-site capture and identification of CPKP are feasible using the ANCA method in combination with the 3D complex nanopillar array swab.

## Discussion

We developed the ANCA method by incorporating an artificial nucleic acid circuit that fully exploits both the target recognition ability and the cleavage activities of Ago proteins. Artificial biochemical circuits have the potential to precisely control the dynamics and operation of chemical reaction networks, which is promising for biosensing, bioregulation, and bioengineering applications[53,54]. In particular, catalytic nucleic acid circuits combine cascaded or cyclic processes to generate multiple outputs in response to a single input, providing valuable toolkits for signal amplification[55]. These circuits facilitate autonomous isothermal amplification with distinct properties that make them suitable for point-of-care and in-field medical diagnostic testing. Despite the construction of numerous catalytic nucleic acid circuits in previous research[56–59], including polymerase/exonuclease/nickase DNA circuits, self/cross-catalytic nucleic acids, hybridization chain reactions, catalytic hairpin assemblies, etc., there have been no reported instances of nucleic acid circuits utilizing Ago proteins. The ANCA is a positive feedback circuit consisting solely of DNA and Ago proteins. Upon introduction of a single target DNA as an input, this circuit generates an exponential fluorescence signal as an output. Furthermore, the ANCA method does not require a PAM sequence for target recognition using Ago proteins. This expands the range of nucleic acid targets that can be detected and increases its versatility in diagnostic applications.

The escalating prevalence of CPE has raised significant public health concerns. As recorded by WHONET, the incidence of CPE acquisition has noticeably escalated from 0.21 cases/1000 patient days (PD) in 2015 to 1.89 cases/1000 PD in 2019[60]. The surge in antibiotic use, driven by the emergence of SARS-CoV-2, predicts a continued increase in CPE prevalence[61]. Hence, early detection of CPE becomes a critical measure to regulate the uncontrolled use of antibiotics and prevent further spread of infection. Currently, most hospitals use the Cepheid Xpert Carba-R assay to identify CPE[62], which is effective but requires specialized equipment for purification and PCR. We have developed the ANCA method, an on-site, user-friendly approach to CPE detection. Unlike PCR, the ANCA method performs all reactions isothermally without purification, simplifying the procedure and reducing the need for extensive equipment. During the repeated lysis and washing process for nucleic acid extraction, some nucleic acids are inevitably lost. In the ANCA method, rectal swabs and transport media are used directly, allowing the detection of DNA without loss. We further evaluated the functionality of the ANCA method using a commercially available portable isothermal nucleic acid amplification device (Supplementary Fig. 18). After testing a total of 20 clinical samples (10 positive for KPC and 10 negative), the ANCA method accurately diagnosed the samples. This finding underscores the versatility of the ANCA method in various experimental setups and its potential for on-site CPE detection compared to previous experiments (Supplementary Table 3).

The integration of the ANCA method with the 3D nanopillar array swab demonstrates the potential to improve the efficiency of CPKP detection. The distinct properties of the 3D nanostructures facilitate irreversible capture of harmful bacteria, thereby promoting safe diagnostic procedures. The 3D nanopillar array swab was fabricated by replicating a nanopillar array and growing polyaniline nanofibers on the array[52]. The swab was shown to effectively capture bacteria in a quantitative range of $10^2$ to $10^7$ CFU/mL[63], with a maximum entrapment coverage of 75 bacteria/100 µm². The high capture efficiency and extensive entrapment coverage can improve the sensitivity of bacterial detection, and potentially, the accuracy of diagnosis. The integration of the ANCA method with the 3D nanopillar array swab thus signifies a promising advancement in diagnostic tools for CPE and other antibiotic-resistant bacteria.

Impressively, the ANCA method has enabled the direct detection of CPKP from patient-collected rectal swabs. Furthermore, the diagnostic results obtained from rectal swabs proved to be more accurate than those obtained from swab-immersed transport media. Imagining a scenario where clinical samples are procured using the 3D nanopillar array swabs and directly subjected to the ANCA method, one could anticipate an accurate diagnosis of CPKP without the risk of secondary infection, as such an approach can circumvent the potential for sample contamination and limit the spread of bacteria. Moving forward, the efficacy of the ANCA method in conjunction with 3D nanopillar array swabs will be carefully evaluated. This will involve close collaboration with physicians to improve our understanding and implementation of this method in real-world clinical settings.

The ANCA method represents a pioneering attempt to synergize artificial nucleic acid circuits with Ago proteins. Consequently, its relatively low LOD can be attributed to the early stage of experimental optimization, especially when compared with established modalities for the detection of pathogenic microorganisms. Future avenues of research may explore enhancing the cleavage efficiency of the Ago protein/guide DNA through an iterative accumulation of experimental knowledge. Nevertheless, an evaluation of the $C_t$ values associated with the clinical samples studied in this investigation reveals parallels with $C_t$ values documented in the contemporary clinical literature[64–66]. Such congruence suggests that the ANCA technique has adequate robustness for clinical applications. The relatively long reaction time compared to other technologies is a drawback of ANCA. This is inevitable to inhibit the apo-form of the Ago protein. The apo-form is often formed in the absence of guide DNA, causing a non-specific cleavage reaction of dsDNA. To inhibit this, the concentration of Ago protein should not be too high. However, due to the nature of the ANCA technique, the higher the concentration of Ago protein, the more favorable the reaction kinetics. Therefore, it is necessary to choose a concentration condition that moderately addresses both issues, which led us to choose 200 nM Ago protein. However, we are hopeful that this issue will be resolved in the near future. This expectation is based on recent advances in protein engineering, such as inactivated Cas proteins. As research into the function of each domain of the Cas9 nuclease continues, Cas9 nucleases that cut only one strand or inactivated Cas9 nucleases with inhibited cleavage activity have emerged. Given the increased interest in Ago protein research, we anticipate that an Ago protein with an inhibited apo-form will be available. Even in the absence of such protein engineering, the reaction rate can be increased by incorporating additional amplification-type circuitry, such as entropy-driven circuitry or catalytic hairpin assembly. Another drawback is that the data presented in the current NEB guide to DNA design is highly variable. This would explain the difference in sensitivity values between each circuit of the ANCA technology. As more data is collected, it may be possible to detect different target nucleic acids with consistent efficiency.

## Methods
### Ethical statement
The methodology adopted for this study was rigorously reviewed and approved by the Institutional Review Board at Gyeongsang National University Changwon Hospital, Changwon, Korea (IRB approval number 2022-10-012). This research complies with all relevant ethical regulations. All participants gave their written informed consent for research use.

**Materials.** All oligonucleotides used in this study were purchased from Integrated DNA Technology (Coralville, IA, USA) and their sequences are listed in Supplementary Table 1. *Thermus thermophilus* argonaute (TtAgo) and Monarch® Plasmid Miniprep Kit were purchased from NEB

(Beverly, MA, USA). Gel-Red Nucleic Acid Stain (41003) was purchased from Biotium (Fremont, CA, USA). Both 6× DNA Loading Buffer and SYBR Green I Nucleic Acid Gel Stain were purchased from Takara Korea Biomedical Inc. (Seoul, Korea). The TOPsimple preMIX (aliquot)-nTaq kit was purchased from Enzynomics (Daejeon, Korea). Human urine and blood samples were purchased from Innovative Research (Novi, MI, USA). Finally, transport media were purchased from Asanpharm (Seoul, Korea). The AccuPrep® Stool DNA Extraction Kit was purchased from Bioneer (Daejeon, Korea).

**Evaluation of ANCA method.** The target DNA for KPC (2 μL) was combined with a solution (18 μL) containing reaction buffer (20 mM of Tris-HCl, 10 mM of $(NH_4)_2SO_4$, 10 mM of KCl, 10 mM of $MgCl_2$, 0.1% Triton X-100, pH 8.8), 200 nM of TtAgo, 25 nM each of G1 and G2, 500 nM each of R and R*, 75 mM of NaCl, and 10 μg/mL of BSA. This mixture was then incubated at 75 °C, and the fluorescence signal was monitored at 1-min intervals using a CFX Opus 96 RT-PCR system (Bio-Rad, Hercules, CA, USA). The ANCA reaction without R* was performed under the same conditions but without the presence of R*.

For PAGE analysis, the target DNA (2 μL) was combined with the ANCA reaction mixture (18 μL) and incubated at 75 °C for 70 min (in the case of Fig. 2d, the incubation time was 400 min). The reaction product (10 μL) was then combined with 6× DNA loading buffer (2 μL) and loaded onto a 15% polyacrylamide gel. After electrophoresis, the gel was stained with Gel-Red Nucleic Acid staining dye and visualized using the Geldoc Go Imaging system (Bio-Rad).

**Direct detection of antibiotic-resistant bacteria using ANCA method.** All bacteria clinical isolates were provided by Gyeongsang National University College of Medicine. The bacteria strains were cultured on Luria-Bertani (LB) agar medium at 37 °C for 16 h, then harvested and resuspended in sterilized phosphate-buffered saline (PBS). The optical densities ($OD_{600}$) of the suspensions were determined at 600 nm using a Nanophotometer P330 (Implen, Germany) to facilitate bacterial count estimation.

To confirm the presence of CPKP-induced genes in the bacteria, gene-specific primers were designed for each KPC and IMP gene using the Primer 3 tool (http://primer3.wi.mit.edu) and the NCBI Reference Sequence Database (https://www.ncbi.nlm.nih.gov/nuccore/ GCF_000240185.1 (WT), NG_049244.1 (KPC), MH909334 (IMP), AAY33963.1 (VIM), AFN84620.1 (NDM), JN626286.1 (OXA-48)). For qRT-PCR, DNA was extracted from antibiotic-resistant bacteria using the Monarch® Plasmid Miniprep Kit. Extracted DNA (2 μL) was combined with a PCR reaction mixture (18 μL) containing the gene-specific primer set (0.5 μM each), 1× TOPsimple preMIX (aliquot)-nTaq, and 1× SYBR Green I dye. Amplification was then proceeded with the following steps: 49 cycles of denaturation for 30 s at 95 °C, annealing for 1 min at 55 °C, and extension for 1 min at 72 °C. The fluorescence signal from the PCR was monitored at each extension step using a CFX Opus 96 RT-PCR system.

For direct detection of antibiotic-resistant bacteria using the ANCA method, $10^5$ cells/mL of bacteria (2 μL) were combined with the ANCA reaction components (18 μL) and incubated following the aforementioned protocol. Each bacterial sample was separately mixed with KPC, IMP, VIM, NDM, and OXA-48 circuit mixtures.

To verify the applicability of the ANCA method for detecting bacteria in clinical specimens, each bacterial strain was spiked into human urine and blood samples ($10^5$ cells/mL). In detail, the 99% bacteria-spiked human sample was prepared by mixing 1 μL of bacteria at a concentration of $10^7$ cells/mL with 99 μL of human sample, and the 90% bacteria-spiked human sample was prepared by mixing 90 μL of human sample with 10 μL of bacteria at a concentration of $10^6$ cells/mL. Prior to applying the ANCA method, qRT-PCR was performed to confirm the presence of CPKP-induced gene in the bacteria-spiked human sample. DNA was extracted using the

Monarch® Plasmid Miniprep Kit and PCR was performed following the aforementioned protocol.

Additionally, instead of using target DNA, distilled water, urine, and blood (2 μL) were combined with KPC circuit mixture (18 μL) to determine the robustness of the experiment against autofluorescence from human urine and blood samples. Next, the bacteria-spiked urine and blood samples (2 μL) were combined with KPC and IMP circuit mixtures (18 μL) and then subjected to the same protocol described above.

**Diagnosis of antibiotic-resistant bacteria using ANCA method.** A total of 143 clinical samples (63 positive for KPC and 80 negative) were acquired from Gyeongsang National University College of Medicine. Rectal swab samples were collected from patients using cotton swabs, subsequently stored in transport media (5 mL), and preserved at −80 °C until further use.

For the diagnosis of antibiotic-resistant bacteria using the ANCA method, the rectal swab-immersed transport media (2 μL) was combined with the ANCA reaction components (18 μL) and incubated following the aforementioned protocol.

For the comparison of ANCA method with qRT-PCR, DNA was extracted from the rectal swab-immersed transport media using the AccuPrep® Stool DNA Extraction Kit. The extracted DNA (2 μL) was combined with PCR reaction mixture (18 μL) containing the KPC gene-specific primer set (0.5 μM each), 1× TOPsimple preMIX (aliquot)-nTaq, and 1× SYBR Green I dye. The fluorescence signal at each extension step was recorded using the CFX Opus Real-Time System (Bio-Rad). The accompanying software (CFX Maestro) was used to obtain $C_t$ values.

To prepare a mock clinical sample for *K. pneu* (KPC), the rectal swabs in transport media, negatively diagnosed by the hospital, were randomly selected. These swabs were then randomly sprayed with *K. pneu* (KPC) at various concentrations and immediately immersed in transport media. Finally, the mock rectal swab-immersed transport media (2 μL) were combined with the ANCA reaction components and incubated following the aforementioned protocol.

To prepare a mock clinical sample for *K. pneu* (IMP), the rectal swab-immersed transport media (500 μL each), negatively diagnosed by the hospital, were randomly collected. Subsequently, *K. pneu* (IMP) at various concentrations was randomly added to the transport media. Finally, the mock rectal swab-immersed transport media (2 μL) were combined with the ANCA reaction components and incubated following the aforementioned protocol.

For the diagnosis of antibiotic-resistant bacteria using the ANCA method, rectal swabs collected from patients and mock rectal swabs for *K. pneu* (IMP) were immersed in distilled water (500 μL). Subsequently, the rectal swab-immersed water (2 μL) was combined with the ANCA reaction components (18 μL) and incubated following the aforementioned protocol.

The portable isothermal nucleic acid amplification device was manufactured by Revosketch (Daejeon, Korea). Twenty rectal swab-immersed transport media (2 μL each) were combined with the ANCA reaction components and incubated following the aforementioned protocol.

**Capture and detection of antibiotic-resistant bacteria using ANCA method with 3D nanopillar swab.** The 3D nanopillar array swabs were prepared as described previously[50,61]. Briefly, Si wafers underwent oxidation in a furnace (Furnace E1200, Centrotherm, Germany) to form $SiO_2$ layers and were then coated with 0.7 μm of photoresist for patterning. Using a KrF scanner (S203−B, Nikon, Japan), 500 nm dots were patterned, followed by inductively coupled plasma (ICP, TCP9400SE, Lam Research, USA) etching to create Si nanoholes. These wafers served as molds for the nanopillar arrays, made by spin coating a polyurethane (MINS-311RM, Minuta Tech., Korea) and NOA 63

(Norland Optical Adhesives, USA) mixture and then pressing a polyethylene terephthalate film (Mitsubishi, Japan) onto it. UV exposure (EVG6200, EVG, Austria) solidified the structure, which was detached from the Si mold. Ti/Au layers were evaporated onto the nanopillars, which were then treated in a chemical solution to grow nanonetworks. After rinsing, these nanostructured films were fixed onto a 3D printed swab backbone, completing the assembly of a 3D nanostructure. Each bacterium ($10^5$ cells/mL) was dropped onto a micropig Franz cell membrane (APURES, Seoul, Korea), desk, scissors, glove, knob, and tweezers. Bacteria were collected from the contaminated surfaces using 3D nanopillar swabs by simple touching and rubbing. SEM images were taken using a Nova 230 system (FEI, Hillsboro, OR, USA) at an accelerating voltage of 15 keV. The piece of bacteria-captured 3D nanopillar swab was mixed with the ANCA reaction components (18 µL), and distilled water (2 µL) was added to make the final volume of 20 µL. The entire mixture was then incubated following the aforementioned protocol.

### Statistics and reproducibility

The sample size was determined according to the Korean Food & Drug Administration. Since the clinical sensitivity and specificity of molecular diagnostics approved by the Korean Food & Drug Administration (KFDA) are 95% each, it is necessary to test at least 80 positive samples using the ANCA method. No data were excluded from the analyses. Samples are randomly allocated. The investigators were blinded during data acquisition and analysis of clinical samples.

### Reporting summary

Further information on research design is available in the Nature Portfolio Reporting Summary linked to this article.

## Data availability

NCBI Reference Sequence Database was utilized for the primer and probe design. (GCF_000240185.1, NG_049244.1, MH909334 [https://www.ncbi.nlm.nih.gov/nuccore/MH909334.1/], AJ870988.1, OL348378.1, and JN626286.1. Source data are provided with this paper.

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

## Acknowledgements

This research was supported by NRF and NST grants funded by Korea government (MSIT) (NRF-2021M3E5E3080379 to T.K., NRF-2021M3H4A1A02051048 to T.K., NRF-2023R1A2C2005185 to T.K., NRF-2021M3E5E3080844 to J.J., NRF-2022R1C1C1008815 to E.-K.L. and CPS22021-100 to E.-K.L.), Technology Development Program for Biological Hazards Management in Indoor Air through KEITI funded by Korea government (ME) (2021003370003 to T.K.), KEIT grant funded by Korea government (MOTIE) (RS-2022-00154853 to T.K.), Nanomedical Devices Development Program of National Nano Fab Center, and KRIBB Research Initiative Program (KGM5472322 to T.K.).

## Author contributions

T.K. supervised the project and wrote the manuscript. H.J. conceived and conducted the experiments, analyzed the data, and wrote the manuscript. J.S. conceived the experiments and analyzed the data. S.K., J.-H.B., K.G.L., K.-H.P., E.W., E.-K.L. and J.J. provided materials and assistance. All authors edited the manuscript.

## Competing interests

The authors declare no competing interests.
