## [Peer Review File · Nature Communications]

Reviewers' comments:

Reviewer #1 (Remarks to the Author):

Remarks to the Authors:

In this manuscript, authors present a pAgo-based DNA detection method, named ANCA. The key component of the method is a DNA-guided Argonaute nuclease (TtAgo), which can use short DNA as guide to cleave complementary DNA with high precision and specificity. They utilize TtAgo with a pair designed DNA guides to cleave double-strand target DNA, resulting in the release of a short DNA that serves as a new guide for cleaving a rationally designed DNA complex. The DNA complex has self-reporting capabilities, effectively combining target recognition and cleavage activity of the Ago system. ANCA establishes a positive feedback loop. The advantages of the ANCA system include – only a protein required, exponential signal amplification, without nucleic acid extraction step and amplification steps, and one-pot reaction. However, there are some issues with this manuscript.

Comments

1.(line 87, page 4) In recent years, there have been numerous publications about Ago-based nucleic acid detection methods. Some publications are also described in the discussion section by authors. Hence, it is not appropriate for Ago proteins to have the potential for application in molecular diagnosis. And, this paragraph lacks an introduction to the existing nucleic acid detection methods based on Ago protein and the advantages of Ago protein for nucleic acid detection.

2.(line 174, page 8) In the figure 2C and 2D, the marks are not aligned with the lanes.

3.(line 201, page 9) The sensitivity of the ANCA method for KPC and IMP was 18.2 fM and 2.5 fM, respectively. The sensitivity is much lower than the other detection method of pathogenic microorganisms, so the authors should explain whether this level of sensitivity meets the standard for routine use.

4.(line 421, page 21) Wang, F et al. integrated RT-PCR with Ago proteins to detect SARS-Cov2. This needs appropriate modification.

5.The discussion section is too long. Some descriptions of the Ago-based detection methods can be described in the introduction.

Reviewer #2 (Remarks to the Author):

The escalating prevalence of CPE has raised significant public health concerns. The authors reported a novel strategy for detecting CPE using ANCA method. This strategy could contribute to simple, rapid, and accurate CPE diagnosis. However, if some problems cannot be explained or solved well, it is not recommended to accept this manuscript.

1. Although carbapenemase-producing *Klebsiella pneumoniae* is the major carbapenemase-producing Enterobacteriaceae strain in most countries and regions, CPKP is not equivalent to CPE. Some sections in this paper used ANCA method to detect CPKP, but CPKP was replaced with CPE in corresponding conclusion, which is not rigorous.

2. In some experiments, the authors did not use other conventional or advanced methodologies to compare with ANCA method, resulting in the potential advantages of this methodology (such as rapid and accurate) not being well highlighted. This is not advisable in methodology-research articles. It is recommended that the authors add more methods to the current experimental framework, and highlight the advantages of ANCA method through intuitive comparison.

3. Line 386-387. Why choose “micropig skin, desk, glove, and scissors” as research subjects here? Random? Is there any special standard?

4. Please describe the disadvantages of ANCA method used in this paper in the discussion section.

Reviewer #3 (Remarks to the Author):

Hyowon Jang et al. reported a single-enzyme, amplification-free, and isothermal detection of nucleic acids by combining artificial nucleic acid circuit with Argonaute (Ago) protein (ANCA). Using the ANCA assay, they detected the Carbapenemase-producing Enterobacteriaceae (CPE), including *Klebsiella pneumoniae* carbapenemase (KPC) and Imipenemase (IMP). Especially, they demonstrated that the ANCA assay could be directly employed to detect nucleic acids in human blood and urine, without need for nucleic acids extraction and amplification steps. Further, they validated the clinical utility of the ANCA assay by detecting both KPC and IMP in clinical samples, achieving a comparable performance with conventional PCR method. Although different catalytic nucleic acid circuits (including CRISPR-based catalytic nucleic acid circuit) have been designed and constructed for nucleic acid detection in previous literature, it is potentially novel and interesting to utilize Ago proteins to develop catalytic nucleic acid circuits for signal amplification detection of nucleic acids. However, I do have some concerns and comments below.

Specific comments:

1. Figure 1 needs to be re-designed because it is similar to Figure 2a. In addition, it does not clearly explain the working principle of the ANCA assay. For instance, in the main text, the author mentioned that the 5' end of the guide DNA plays an important role in binding to the Ago protein. However, there is no indication of their 5' end and 3' end in Figure 1.
2. There are several concerns in Figure 2: i) even if ANCA assay is referred to as a positive feedback circuit, it seems that fluorescence signals should be observed even with only R present. Why is there no signal detected at all (Fig. 2d)? ii) more control groups are needed to evaluate the ANCA method. For example, there is no result to show the cleavage of the target sequence by the Ago/G1 and Ago/G2 or either one. It is unclear whether the T1 can be produced or released by the cleavage reaction between Ago and the target. iii) The experimental explanation in the text and the figure are very vague and unclear. Does Ago protein mean Ago /G1 and Ago/G2 in Figure 2c and d?
3. In Figure 4, the authors demonstrated that the bacteria in raw samples (e.g., urine, blood) can be directly detected without DNA purification, how did they address the issue of autofluorescence (or background) caused by raw samples themselves (e.g., especially for blood in Figure 4g)?
4. They stated that the 3D nanopillar array structure enables the irreversible entrapment of bacteria. In the method section, it was mentioned that the bacteria-captured nanopillar structure was immersed in distilled water. However, it does not make sense that irreversibly entrapped bacteria can be easily separated from the nano-structure by simply soaking. How can the captured bacteria be released for analysis?
5. In the ANCA method, its limit of detection (LoD) of fM (18.2 fM for KPC and 2.5 fM for IMP) are not very impressive. Its LOD may be lower when testing clinical samples. Typically, PCR method can achieve a LoD of aM when testing extracted or purified nucleic acid. However, in their clinical validation, it seems that the extraction-free ANCA method could achieve a comparable performance with PCR method (Figure 5). Did the authors extract DNA before running PCR testing? If so, how could their extraction-free, amplification-free ANCA method achieve a sensitivity of 100% and specificity of 100% compared with PCR method (including nucleic acid extraction and enzymatic amplification)?
6. The sequence of oligonucleotide T1 for KPC or IMP should be provided in the supplementary table 1. In addition, what is the reference gene used for PCR?
7. The authors should explain how the false positive signals were generated in their ANCA method. Why do their false positive curves have still a sigmoidal shape which is typically caused due to exponential amplification?
8. In line 290-292, they mentioned that "A strong correlation was observed between the T₀ – T and Ct₀ – Ct values for KPC-positive clinical samples (Fig. 5c). Ct₀ was set to 50 and Ct denotes the cycle threshold of PCR for clinical sample." T₀ represents the threshold time for negative control sample or threshold time for the pure transport media. However, why did they set Ct₀ to 50, not the threshold time of the pure transport like T₀. In addition, in supplementary Figure 3, the cycle threshold of the negative sample is less 50.

9. The experimental section should be written in more detail. For example, how was the bacteria concentration 10^5 cells/mL estimated? And how were they spiked in human samples?

10. What is the limitation of this study? The drawbacks of the study are needed to be discussed.

We appreciate the editor and reviewers for their valuable comments to improve our manuscript. The changes in the manuscript and the responses to the editor and reviewers' comments are as follows:

Reply to Reviewer 1

(Comment) In this manuscript, authors present a pAgo-based DNA detection method, named ANCA. The key component of the method is a DNA-guided Argonaute nuclease (TtAgo), which can use short DNA as guide to cleave complementary DNA with high precision and specificity. They utilize TtAgo with a pair designed DNA guides to cleave double-strand target DNA, resulting in the release of a short DNA that serves as a new guide for cleaving a rationally designed DNA complex. The DNA complex has self-reporting capabilities, effectively combining target recognition and cleavage activity of the Ago system. ANCA establishes a positive feedback loop. The advantages of the ANCA system include – only a protein required, exponential signal amplification, without nucleic acid extraction step and amplification steps, and one-pot reaction. However, there are some issues with this manuscript.

Response) Thank you for taking the time to review our manuscript. We appreciate your recognition of the strengths of the method, including its high precision, specificity, and the advantages of requiring only one protein for exponential signal amplification in a one-pot reaction. We understand that you have identified some problems with our manuscript. As authors committed to scientific rigor and quality, we are eager to fully address these concerns in a revised submission. Thank you again for your valuable time and feedback.

(Question 1) (line 87, page 4) In recent years, there have been numerous publications about Ago-based nucleic acid detection methods. Some publications are also described in the discussion section by authors. Hence, it is not appropriate for Ago proteins to have the potential for application in molecular diagnosis. And, this paragraph lacks an introduction to the existing nucleic acid detection methods based on Ago protein and the advantages of Ago protein for nucleic acid detection.

Response) Thank you for your insightful comments, particularly regarding the positioning of Ago proteins in the field of molecular diagnostics and the inadequate description of existing methods based on Ago proteins.

Upon reflection, we agree that the original wording did not adequately capture the role of Ago proteins in molecular diagnostics. Accordingly, we have revised the manuscript to state that their stringent DNA-guided target recognition capabilities and unique cleavage activities have made Ago proteins remarkable in bioanalysis. This adjustment should more accurately reflect the current state of the field.

We also recognize the merit of your suggestion to elaborate on existing Ago-based methods and the advantages of Ago protein for nucleic acid detection. To address this, we have consulted a recent review article on Ago protein-based diagnostic techniques and incorporated pertinent information into our manuscript. In the interest of clarity and comprehensiveness, we have moved this information from the Discussion section to the Introduction section.

With these revisions, we believe that we have significantly strengthened the manuscript and hope that we have adequately addressed your concerns. We are grateful for your constructive feedback, which has undoubtedly helped to improve the quality of our work.

[Introduction]

Argonaute (Ago) proteins, named for their homology to the Ago family of proteins in eukaryotes, play a critical role in RNA interference (RNAi) and small RNA-guided gene regulatory pathways¹⁹. These proteins are found in a wide range of organisms^{19, 20}, including eukaryotes, prokaryotes and archaea, underscoring their evolutionary conservation and functional importance²¹. Their stringent DNA-guided target recognition capabilities and unique cleavage activities have made Ago proteins remarkable in bioanalysis. One outstanding feature is their ability to cleave target nucleic acids without requiring a specific sequence motif, unlike the protospacer adjacent motif (PAM) used by Cas proteins^{22, 23}. Instead, Ago proteins achieve precise cleavage by base-pairing with a segment of guide DNA between positions 10 and 11^{24, 25}, offering adaptability with less restrictive nucleic acid selection. Thermophilic variants of these proteins, derived from heat-loving microorganisms²⁶, add another layer of attractiveness due to their ability to withstand high temperatures, enhancing the robustness of Ago-based diagnostic methods. In addition, the use of a DNA-based guide probe, which is more stable and less expensive than RNA, can further increase the effectiveness of the assay.

In current research, Ago proteins are mainly used for direct cleavage after nucleic acid amplification reactions such as PCR, loop-mediated isothermal amplification (LAMP), etc. For example, He, R *et al.*²⁷, Song, J *et al.*²⁸, and Liu, Q *et al.*²⁹ used PCR to amplify specific target DNAs, which were then cleaved by Ago proteins, producing an enhanced fluorescence signal. Wang, F *et al.*³⁰ used reverse transcription (RT)-PCR to convert severe acute respiratory syndrome coronavirus 2 (SARS-CoV-2) RNA into cDNA. This cDNA was then subjected to two cleavage reactions triggered by Ago proteins, resulting in a fluorescent signal. Ye, X *et al.*³¹ used LAMP technology to generate cDNA from SARS-CoV-2 and influenza RNA, followed by cleavage reactions similar to the method of Wang, F *et al.* Li, Y *et al.*³² combined exonuclease I with Ago proteins, amplified food poisoning bacterial DNA by PCR, and then generated a fluorescent signal by a cleavage reaction. In contrast, Yuan, C *et al.*³³ used Ago proteins for initial nucleic acid cleavage prior to pre-amplification. The resulting RNA fragment was then subjected to an exponential amplification reaction with dsDNA identification *via* SYBR I dye. Wang, L *et al.*³⁴ combined the ligation chain reaction with Ago proteins. This reaction joined two short strands, directed cleavage of the reporter probe, and produced a fluorescent result. Although these methods showed promising detection capabilities by utilizing the superior target recognition capabilities of Ago proteins, they require an increased number of enzymes and procedural steps, resulting in a relatively complex process. Meanwhile, Shin, S *et al.*³⁵ and Li, X *et al.*³⁶ devised methods using Ago proteins without nucleic acid amplification, however, relatively poor performances were recorded.

(Question 2) (line 174, page 8) In the figure 2c and 2d, the marks are not aligned with the lanes.

Response) Thank you for bringing to our attention the misalignment of marks with lanes in Figure 2c and d. We have corrected this issue to ensure that the marks are now properly aligned with their corresponding lanes. Once again, we are grateful for your careful review, which has helped to improve the quality of our manuscript.

[Figure]

Fig. 2 Evaluation of ANCA method. (a, b) Schematic illustration of (a) ANCA and (b) ANCA without R*. ANCA method enables exponential amplification through positive feedback. ANCA without R* cleaves target DNA and R, but no further reaction transpires. (c) Time-dependent fluorescence intensities during ANCA reaction (left) and corresponding PAGE analysis result (right) under various components. (d) Time-dependent fluorescence intensities during ANCA reaction without R* (left) and corresponding PAGE analysis result (right) under various components. [Target DNA] = 1 nM, [R] = 500 nM, [R*] = 500 nM, [G1] = 25 nM, [G2] = 25 nM, [Ago] = 200 nM, [MgCl₂] = 10 mM, [NaCl] = 75 mM, and [BSA] = 10 μg/mL.

(Question 3) (line 201, page 9) The sensitivity of the ANCA method for KPC and IMP was 18.2 fM and 2.5 fM, respectively. The sensitivity is much lower than the other detection method of pathogenic microorganisms, so the authors should explain whether this level of sensitivity meets the standard for routine use.

Response) We appreciate the reviewer's astute observation regarding the sensitivity of our ANCA method. While it's true that our originally reported LOD values were relatively low compared to other conventional methods for the detection of pathogenic microorganisms, we took corrective action after submission.

During further investigation, we identified an influential component within the enzyme storage buffer, distinct from the reaction buffer, that was affecting performance. By optimizing the concentration of this component (Supplementary Figure 8), along with other critical components, we were able to significantly improve the sensitivity of our method. The resulting LOD values for the KPC and IMP circuits were 1.87 fM and 178 aM, respectively, as shown in revised Figure 3.

In addition, we have successfully extended our work to the detection of VIM, NDM and OXA-48, thereby enhancing the clinical utility of the ANCA technology (Supplementary Figure 9 and 12). The inclusion of these genes, along with KPC and IMP, is critical because these five genes are integral to CPE as they are most often responsible for conferring resistance to carbapenems.

We have created an additional comparison table that compares the sensitivity of the ANCA method to other established techniques. Interestingly, the sensitivity of ANCA was found to be superior when compared to other Ago protein-based detection methods without pre-amplification. This finding highlights the potential scalability and improvement opportunities for ANCA within the field of Ago protein-based techniques.

It's important to note that the revised sensitivity levels are well aligned with the clinical context. As shown in Figure 5, the ANCA method has demonstrated its ability to detect clinically relevant specimens, which showed C_t values ranging from 12 to 33, mirror the C_t values reported in established medical research. These parallels underscore the ability of ANCA to effectively identify typical clinical specimens.

In light of these findings, we believe that ANCA sensitivity remains relevant in the clinical setting. In essence, the ability to detect authentic KPCs and IMPs does not require excessive sensitivity. An overemphasis on ultra-high sensitivity could inadvertently increase false-positive rates, leading to unwarranted clinical concerns.

We have included the aforementioned performance comparison table in the Supplementary Information and made the necessary changes to the manuscript to reflect these clarifications.

Thank you again for your valuable feedback, which has helped us to improve the quality of our work.

[Results]

Optimization and sensitivity of ANCA method

To determine the optimal conditions for the ANCA method, we first monitored changes in fluorescence by varying R and R* (Supplementary Fig. 7). When the fluorophore and quencher were attached to the R sequence only, the ANCA method demonstrated efficient target DNA detection. It is worth noting that when the fluorophore and quencher were attached to both R and R*, the reaction efficiency was significantly reduced, which may be attributed to steric hindrance affecting the cleavage activity of Ago/guide DNA complexes. We then optimized the reaction temperature and the concentrations of R, R*, Ago protein, and Mg²⁺ for the ANCA method. As shown in Supplementary Fig. 8a-e, the conditions for optimal detection performance were found to be a reaction temperature of 75 °C and concentrations of 500 nM for R and R*, 200 nM for Ago protein, and 10 mM for MgCl₂. In addition, the concentrations of NaCl and bovine serum albumin (BSA) in the enzyme storage buffer were optimized to 75 mM and 10 µg/mL, respectively (Supplementary Fig. 8f and g). These conditions were used in subsequent experiments.

Under the optimized conditions, we evaluated the sensitivity of the ANCA method for KPC and IMP as target sequences, respectively. Fig. 3a shows the time-dependent fluorescence intensities for different concentrations of the KPC sequence. The threshold time was determined as the reaction time at which the fluorescence intensity reached 10,000. Fig. 3b shows the plot of threshold time as a function of target DNA concentration. A linear relationship ($R^2 = 0.99$) was observed in the range of 10 fM to 10 nM, and the limit of detection (LOD) was calculated to be 1.87 fM using the formula³⁹ (limit of blank (LOB) = mean of blank + 1.645 × standard deviation of blank, LOD = LOB + 1.645 × standard deviation of low concentration sample). Additionally, we adapted the ANCA method for the detection of IMP. Using the IMP circuit, a linear relationship ($R^2 = 0.99$) was observed in the range of 1 fM to 1 nM, and the LOD was calculated to be 178 aM (Fig. 3c and d). Similarly, circuits for the detection of VIM, NDM, and OXA-48 were fabricated and evaluated for their sensitivity to each of the target substances. In all cases, a linear relationship ($R^2 = 0.99$) was observed in the range of 1 fM to 1 nM, and the LOD of VIM, NDM, and OXA-48 were calculated to be 529, 120, and 144 aM,

respectively (Supplementary Fig. 9). The sensitivity of the ANCA method is far superior to previous results that utilize Ago proteins without pre-amplification (Supplementary Table 2)²⁷⁻³⁶. Furthermore, it is noteworthy that the ANCA method demonstrates flexibility and broad target applicability, as evidenced by the successful detection of KPC, IMP, VIM, NDM, and OXA-48 sequences by slight modifications of the ANCA method.

Direct detection of antibiotic-resistant bacteria using ANCA method

We also attempted to detect CPKP using the VIM, NDM, and OXA-48 circuits and found that each circuit was able to discriminate well between WT and CPKP. (Supplementary Fig. 12a-c).

[Discussion]

The ANCA method represents a pioneering attempt to synergize artificial nucleic acid circuits with Ago proteins. Consequently, its relatively low LOD can be attributed to the early stage of experimental optimization, especially when compared with established modalities for the detection of pathogenic microorganisms. Future avenues of research may explore enhancing the cleavage efficiency of the Ago protein/guide DNA through an iterative accumulation of experimental knowledge. Nevertheless, an evaluation of the C_t values associated with the clinical samples studied in this investigation reveals parallels with C_t values documented in the contemporary clinical literature⁶⁴⁻⁶⁶. Such congruence suggests that the ANCA technique has adequate robustness for clinical applications.

[Figure]

Fig. 3 Sensitivity of ANCA method. (a, c) Time-dependent fluorescence intensities during ANCA method with various concentrations of (a) KPC and (c) IMP sequences. Dashed black threshold lines indicate the reaction time at which the fluorescence intensity reached 10,000 (threshold time). (b, d) Correlation of threshold time to the logarithm of (b) KPC and (d) IMP concentration ($n = 3$, error bar = standard deviation). Red lines are linear fits, indicating LODs of 1.87 fM for KPC and 178 aM for IMP. $[R] = 500$ nM, $[R^*] = 500$ nM, $[G1] = 25$ nM, $[G2] = 25$ nM, $[Ago] = 200$ nM, $[MgCl_2] = 10$ mM, $[NaCl] = 75$ mM, and $[BSA] = 10$ μ g/mL.

[Supplementary Information]

Supplementary Fig. 8 (a-g) Plot of $T_0 - T$ value by varying (a) reaction temperature and concentrations of (b) R, (c) R*, (d) Ago protein, (e) $MgCl_2$, (f) NaCl, and (g) BSA for ANCA method ($n = 3$, error bar = standard deviation, N.D. = threshold time was not determined). When the reaction temperature was 75 °C and the concentrations of R, R*, Ago protein, $MgCl_2$, NaCl, and BSA were 500 nM, 500 nM, 200 nM, 10 mM, 75 mM, and 10 µg/mL, respectively, the highest $T_0 - T$ values were obtained. T_0 represents the threshold time for negative sample, and T denotes the threshold time for positive sample. [Target DNA] = 10 nM.

Supplementary Fig. 9 (a, c, e) Time-dependent fluorescence intensities during ANCA method with various concentrations of (a) VIM, (c) NDM, and (e) OXA-48 sequences. Dashed black threshold lines indicate the reaction time at which the fluorescence intensity reached 10,000 (threshold time). (b, d, f) Correlation of threshold time to the logarithm of (b) VIM, (d) NDM, and (f) OXA-48 concentration ($n = 3$, error bar = standard deviation). Red lines are linear fits, indicating LODs of 529 aM for VIM, 120 aM for NDM, and 144 aM for OXA-48. $[R] = 500$ nM, $[R^*] = 500$ nM, $[G1] = 25$ nM, $[G2] = 25$ nM, $[Ago] = 200$ nM, $[MgCl_2] = 10$ mM, $[NaCl] = 75$ mM, and $[BSA] = 10$ μ g/mL.

Supplementary Fig. 12 (a-c) Plot of $T_0 - T$ value as a function of bacteria in buffer using (a) VIM, (b) NDM, and (c) OXA-48 circuits ($n = 3$, error bar = standard deviation). T_0 represents the threshold time for negative control sample (absence of bacteria), and T denotes the threshold time for test sample (presence of bacteria).

Supplementary Table 2 Comparison of ANCA method with previous Ago-based nucleic acid detection methods.

Method	LOD	Target	Mechanism	Property	Ref
Ago-FISH (Argonaute-based fluorescence in situ hybridization)	1 pM	miRNA (Let7a, c)	Ago-based cleavage	- Low sensitivity	[S1]
MULAN (Multiplex Argonaute-based nucleic acid detection system)	1.6 copies/reaction	SARS-CoV-2 Influenza virus	RT-LAMP + Ago-based cleavage	- Requirement of reverse transcription and nucleic acid amplification steps (Multiple steps) - Cannot be proceeded on isotherm	[S2]
NAVIGATER (Nucleic acid enrichment via DNA guided Argonaute from thermus thermophilus)	1 copy/reaction	KRAS mutation	PCR or RT-LAMP + Ago-based cleavage	- Requirement of nucleic acid amplification step (Multiple steps) - Cannot be proceeded on isotherm	[S3]
A-Star (Ago-directed specific target enrichment and detection)	1 copy/reaction	BRAF & EGFR mutation	PCR + Ago-based cleavage	- Requirement of nucleic acid amplification step (Multiple steps) - Cannot be proceeded on isotherm	[S4]
PLCR (Pfago coupled with modified ligase chain reaction for nucleic acid detection)	10 aM	SARS-CoV-2	Ligase chain reaction + Ago-based cleavage	- Requirement of reverse transcription and nucleic acid amplification steps (Multiple steps) - Cannot be proceeded on isotherm	[S5]
PAND (Pfago-mediated nucleic acid detection)	1.6 aM	KRAS & EGFR mutation	PCR or tHDA + Ago-based cleavage	- Requirement of nucleic acid amplification step (Multiple steps) - Cannot be proceeded on isotherm	[S6]
MAIDEN (Mesophilic Ago-based isothermal detection method)	4 nM	SARS-CoV-2	Reverse transcription + Ago-based cleavage	- Requirement of reverse transcription step (Multiple steps) - Low sensitivity	[S7]
SPOT (Scalable and portable Testing)	0.44 copies/uL	SARS-CoV-2	RT-LAMP + Ago-based cleavage	- Requirement of reverse transcription and nucleic acid amplification steps (Multiple steps) - Cannot be proceeded on isotherm	[S8]

Method	LOD	Target	Mechanism	Property	Ref
TtAgoEAR (TtAgo-based thermostable exponential amplification reaction)	10 aM	SARS-CoV-2	Ago-based cleavage + EXPAR	- Requirement of lots of proteins	[S9]
NOTE-Ago (Novel and One-step cleavage method based on Argonaute by integrating Tag-specific primer extension and Exonuclease I)	1 CFU/mL	S. Typhi S. aureus	PCR + Exo I-based digestion + Ago-based cleavage	- Requirement of nucleic acid amplification step (Multiple steps) - Cannot be proceeded on isotherm	[S10]
ANCA	1.87 fM (KPC) 178 aM (IMP) 529 aM (VIM) 120 aM (NDM) 144 aM (OXA-48)	CPKP	Ago-based cleavage with nucleic acid circuit	- One-step - Amplification-free - Isothermal	This work

Supplementary Table 3 Comparison of ANCA method with previous CPE detection methods.

Method	Sensitivity	LOD	Advantage	Limitation	Ref
MHT (Culture-based Modified Hodge Test)	>69%		- Simple and cost-effective	- Requires pure culture - Long reaction time for culturing - False-positive and false-negative	[S11]
CIM (Carbapenem-inactivation methods)	>90%		- Cover all carbapenemase - Simple and cost-effective	- Requires pure culture - Long reaction time for culturing	[S12]
Colorimetric assay based CarbaNP test and its automated kits	>70%		- Cover all carbapenemase - Simple and cost-effective - Low false-positive rate than MHT	- Hard to detect OXA-48 type producer - Long reaction time for culturing	[S13]
MALDI-TOF	>92%		- Accurate data analysis - High-throughput	- Requires expertise to analyze data - Requirement expensive equipment - Long reaction time for culturing	[S14]
LAMP	>90%		- Rapid and moderate cost - Applicable in limited-resource settings	- Contamination due to amplification of target nucleic acid - Complex primer design - Requirement of DNA extraction	[S15]
Direct detection of rectal swabs by a multiplex lateral flow immunoassay	80-100%		- Simple and cost-effective - Applicable in limited-resource settings	- Poor sensitivity - Long reaction time for culturing	[S16]
SERS combined with chemometric tool	99.8%		- Rapid and moderate cost - High sensitivity	- Poor reproducibility - Requires specific instrument and multivariate data analysis	[S17]
DNA microarray	99.4%	10 ³ copies/μL (~1.66 fM)	- High multiplexing - High sensitivity	- Requirement of expensive equipment and DNA extraction - Long reaction time for culturing	[S18]
RPA + CRISPR/Cas12	99.29%	4.48 fM	- High sensitivity and specificity - Excellent accuracy	- Contamination due to amplification of target nucleic acid - Multiple reaction steps	[S19]

Method	Sensitivity	LOD	Advantage	Limitation	Ref
LAMP + CRISPR/Cas12 + microfluidic chip		30 copies/reaction (~ 0.88 aM)	 - On-site detection - Naked eye detection - No complicated device 	 - Multiple reaction steps - Requirement of DNA extraction 	[S20]
CRISPR/Cas12 + SWV		3.5 fM	 - Excellent selectivity - Simple and cost-effective 	 - Requirement of DNA extraction - Multiple reaction steps 	[S21]
ECL		0.05 copies/ μ L (~0.08 aM)	 - High sensitivity - Amplification-free 	 - Requirement of DNA extraction - Multiple and long reaction steps 	[S22]
ANCA	100%	1.87 fM (KPC) 178 aM (IMP) 529 aM (VIM) 120 aM (NDM) 144 aM (OXA-48)	 - One-step - Amplification-free - Isothermal 	 - Ago-based cleavage with nucleic acid circuit 	This work

(Question 4) (line 421, page 21) Wang, F *et al.* integrated RT-PCR with Ago proteins to detect SARS-CoV-2. This needs appropriate modification.

Response) We appreciate your diligence in identifying the omission related to the work of Wang, F *et al.* on the integration of RT-PCR with Ago proteins for SARS-CoV-2 detection. We apologize for the omission and appreciate your attention to detail. We have amended the manuscript to accurately reflect the contributions of this particular study.

In addition, consistent with your earlier and next comments (1 and 5), we have moved this corrected information to the Introduction section of the paper. This realignment allows us to better contextualize the existing methods based on Ago proteins right at the beginning of our manuscript, thereby providing a more comprehensive background against which our work can be evaluated.

Thank you again for your insightful feedback, which was instrumental in refining our paper.

[Introduction]

Wang, F *et al.*³⁰ used reverse transcription (RT)-PCR to convert severe acute respiratory syndrome coronavirus 2 (SARS-CoV-2) RNA into cDNA. This cDNA was then subjected to two cleavage reactions triggered by Ago proteins, resulting in a fluorescent signal.

(Question 5) The discussion section is too long. Some descriptions of the Ago-based detection methods can be described in the introduction.

Response) Thank you for your valuable suggestions regarding the length of the Discussion section and the placement of descriptions of Ago-based detection methods. We agree with your assessment and have acted accordingly. To address your point and the point raised in your earlier comment 1, we have moved the description of Ago-based detection methods from the Discussion to the Introduction section of the manuscript. This adjustment not only shortens the Discussion, but also enriches the Introduction by providing a more complete and relevant context for our work.

We appreciate your constructive feedback, which was invaluable in improving the quality of our manuscript.

[Introduction]

Argonaute (Ago) proteins, named for their homology to the Ago family of proteins in eukaryotes, play a critical role in RNA interference (RNAi) and small RNA-guided gene regulatory pathways¹⁹. These proteins are found in a wide range of organisms^{19, 20}, including eukaryotes, prokaryotes and archaea, underscoring their evolutionary conservation and functional importance²¹. Their stringent DNA-guided target recognition capabilities and unique cleavage activities have made Ago proteins remarkable in bioanalysis. One outstanding feature is their ability to cleave target nucleic acids without requiring a specific sequence motif, unlike the protospacer adjacent motif (PAM) used by Cas proteins^{22, 23}. Instead, Ago proteins achieve precise cleavage by base-pairing with a segment of guide DNA between positions 10 and 11^{24, 25}, offering adaptability with less restrictive nucleic acid selection. Thermophilic variants of these proteins, derived from heat-loving microorganisms²⁶, add another layer of attractiveness due to their ability to withstand high temperatures, enhancing the robustness of Ago-based diagnostic methods. In addition, the use of a DNA-based guide probe, which is more stable and less expensive than RNA, can further increase the effectiveness of the assay.

In current research, Ago proteins are mainly used for direct cleavage after nucleic acid amplification reactions such as PCR, loop-mediated isothermal amplification (LAMP), etc. For example, He, R *et al.*²⁷, Song, J *et al.*²⁸, and Liu, Q *et al.*²⁹ used PCR to amplify specific target DNAs, which were then cleaved by Ago proteins, producing an enhanced fluorescence signal. Wang, F *et al.*³⁰ used reverse transcription (RT)-PCR to convert severe acute respiratory syndrome coronavirus 2 (SARS-CoV-2) RNA into cDNA. This cDNA was then subjected to two cleavage reactions triggered by Ago proteins, resulting in a fluorescent signal. Ye, X *et al.*³¹ used LAMP technology to generate cDNA from SARS-CoV-2 and influenza RNA, followed by cleavage reactions similar to the method of Wang, F *et al.* Li, Y *et al.*³² combined exonuclease I with Ago proteins, amplified food poisoning bacterial DNA by PCR, and then generated a fluorescent signal by a cleavage reaction. In contrast, Yuan, C *et al.*³³ used Ago proteins for initial nucleic acid cleavage prior to pre-amplification. The resulting RNA fragment was then subjected to an exponential amplification reaction with dsDNA identification *via* SYBR I dye. Wang, L *et al.*³⁴ combined the ligation chain reaction with Ago proteins. This reaction joined two short strands, directed cleavage of the reporter probe, and produced a fluorescent result. Although these methods showed promising detection capabilities by utilizing the superior target recognition capabilities of Ago proteins, they require an

increased number of enzymes and procedural steps, resulting in a relatively complex process. Meanwhile, Shin, S *et al.*³⁵ and Li, X *et al.*³⁶ devised methods using Ago proteins without nucleic acid amplification, however, relatively poor performances were recorded.

Reply to Reviewer 2

(Comment) The escalating prevalence of CPE has raised significant public health concerns. The authors reported a novel strategy for detecting CPE using ANCA method. This strategy could contribute to simple, rapid, and accurate CPE diagnosis. However, if some problems cannot be explained or solved well, it is not recommended to accept this manuscript.

Response) Thank you for recognizing the potential impact of our ANCA method on the pressing issue of CPE prevalence and for your detailed review, which helped identify areas for improvement in our manuscript. We appreciate your constructive criticism and are committed to addressing the issues you've highlighted.

(Question 1) Although carbapenemase-producing *Klebsiella pneumoniae* is the major carbapenemase-producing Enterobacteriaceae strain in most countries and regions, CPKP is not equivalent to CPE. Some sections in this paper used ANCA method to detect CPKP, but CPKP was replaced with CPE in corresponding conclusion, which is not rigorous.

Response) Thank you for bringing this important distinction between carbapenemase-producing *Klebsiella pneumoniae* (CPKP) and carbapenemase-producing Enterobacteriaceae (CPE) to our attention. We acknowledge the lack of rigor in our use of these terms interchangeably and apologize for any confusion this may have caused.

After reviewing the manuscript based on your feedback, we have amended the relevant sections to accurately distinguish between CPKP and CPE. We sincerely appreciate your constructive criticism as it helps us to improve the quality and accuracy of our work.

[Manuscript]

Since the entire manuscript was changed, please refer to the "List of Changes" file for the detailed changes made to the manuscript.

(Question 2) In some experiments, the authors did not use other conventional or advanced methodologies to compare with ANCA method, resulting in the potential advantages of this methodology (such as rapid and accurate) not being well highlighted. This is not advisable in methodology-research articles. It is recommended that the authors add more methods to the current experimental framework, and highlight the advantages of ANCA method through intuitive comparison.

Response) Thank you for your valuable insights regarding the need for comparative assessments between our ANCA method and other existing methodologies. We acknowledge the absence of such comparisons in the original manuscript and recognize that this shortcoming fails to adequately showcase the unique advantages of ANCA.

In the revised manuscript, we have taken your suggestions to heart and included a more comprehensive evaluation. We compared ANCA not only to other Ago-based methods but also to other prevalent technologies used for detecting CPE. These comparisons are summarized in a new table, aimed at providing an intuitive overview of how ANCA stands out in terms of speed, accuracy, and other significant characteristics.

To further address your concerns about the lack of a comprehensive experimental framework, we performed additional comparative experiments. Specifically, we used PCR on bacteria-spiked urine and blood samples to validate our ANCA method. The results were included as supplementary material and discussed in the manuscript to provide a more intuitive comparison of ANCA performance.

In addition, we performed PAGE analyses to include additional markers and control groups. These results, now shown in Figures 2c and d, provide further support for the high amplification efficiency and sensitivity of the ANCA technique. The evidence supports the establishment of a positive feedback loop in ANCA that allows for increased sensitivity in target DNA detection.

We believe these revisions successfully address your concerns and better demonstrate the utility and advantages of the ANCA method. We sincerely thank you for your guidance, which has significantly improved the quality of our work.

[Results]

Evaluation of ANCA method

The ANCA method was designed to provide exponential amplification through a positive feedback loop (Fig. 2a). We first evaluated the ANCA method with different components using KPC as the target. In the presence of all reaction components, including the target, strong fluorescence signals were observed during the ANCA reaction (spectrum 5 in Fig. 2c). In the absence of target DNA, Ago protein, G1, or G2, negligible signals were detected (spectra 1-4

in Fig. 2c). The feasibility of the ANCA method is further confirmed by the results of polyacrylamide gel electrophoresis (PAGE) analysis (right panel in Fig. 2c). The band indicating the intact R-R* structure (lane M2 in Fig. 2c) was completely degraded after the ANCA reaction (lane 5 in Fig. 2c). On the other hand, in the absence of target DNA, Ago protein, G1, or G2, the bands corresponding to the R-R* structure remain intact (lanes 1-4 in Fig. 2c). To evaluate the amplification efficiency of the ANCA method, we performed comparative experiments (Fig. 2b). In the ANCA reaction without R*, the Ago/G1 and Ago/G2 complexes cleave the target DNA, resulting in the formation of the Ago/T1 complex. The Ago/T1 complex then cleaves R, releasing Output and T2. However, no further reaction occurs because of the absence of R*. As a result, no fluorescence signals were observed even after 400 min of reaction, in the presence of the target sequence (spectrum 5 in Fig. 2d). In addition, no signals were detected in the absence of target DNA, Ago protein, G1, or G2 (spectra 1-4 in Fig. 2d). The PAGE results also show the intact R structure bands for all conditions (right panel in Fig. 2d).

Optimization and sensitivity of ANCA method

The sensitivity of the ANCA method is far superior to previous results that utilize Ago proteins without pre-amplification (Supplementary Table 2)²⁷⁻³⁶.

Direct detection of antibiotic-resistant bacteria using ANCA method

To investigate the ANCA method for the detection of bacteria in clinical specimens, we prepared CPKP-spiked human urine and blood samples. These samples were first tested by PCR using a typical DNA extraction kit (Supplementary Fig. 13)

[Discussion]

This finding underscores the versatility of the ANCA method in various experimental setups and its potential for on-site CPE detection compared to previous experiments (Supplementary Table 3).

[Figure]

Fig. 2 Evaluation of ANCA method. (a, b) Schematic illustration of (a) ANCA and (b) ANCA without R*. ANCA method enables exponential amplification through positive feedback. ANCA without R* cleaves target DNA and R, but no further reaction transpires. (c) Time-dependent fluorescence intensities during ANCA reaction (left) and corresponding PAGE analysis result (right) under various components. (d) Time-dependent fluorescence intensities during ANCA reaction without R* (left) and corresponding PAGE analysis result (right) under various components. [Target DNA] = 1 nM, [R] = 500 nM, [R*] = 500 nM, [G1] = 25 nM, [G2] = 25 nM, [Ago] = 200 nM, [MgCl₂] = 10 mM, [NaCl] = 75 mM, and [BSA] = 10 μg/mL.

[Supplementary Information]

Supplementary Fig. 13 (a, b) RT-PCR results of bacteria in urine (99%) using (c) KPC- and (d) IMP-specific primer. (c, d) RT-PCR results of bacteria in blood (99%) using (e) KPC- and (f) IMP-specific primer.

Supplementary Table 2 Comparison of ANCA method with previous Ago-based nucleic acid detection methods.

Method	LOD	Target	Mechanism	Property	Ref
Ago-FISH (Argonaute-based fluorescence in situ hybridization)	1 pM	miRNA (Let7a, c)	Ago-based cleavage	- Low sensitivity	[S1]
MULAN (Multiplex Argonaute-based nucleic acid detection system)	1.6 copies/reaction	SARS-CoV-2 Influenza virus	RT-LAMP + Ago-based cleavage	- Requirement of reverse transcription and nucleic acid amplification steps (Multiple steps) - Cannot be proceeded on isotherm	[S2]
NAVIGATER (Nucleic acid enrichment via DNA guided Argonaute from thermus thermophilus)	1 copy/reaction	KRAS mutation	PCR or RT-LAMP + Ago-based cleavage	- Requirement of nucleic acid amplification step (Multiple steps) - Cannot be proceeded on isotherm	[S3]
A-Star (Ago-directed specific target enrichment and detection)	1 copy/reaction	BRAF & EGFR mutation	PCR + Ago-based cleavage	- Requirement of nucleic acid amplification step (Multiple steps) - Cannot be proceeded on isotherm	[S4]
PLCR (Pfago coupled with modified ligase chain reaction for nucleic acid detection)	10 aM	SARS-CoV-2	Ligase chain reaction + Ago-based cleavage	- Requirement of reverse transcription and nucleic acid amplification steps (Multiple steps) - Cannot be proceeded on isotherm	[S5]
PAND (Pfago-mediated nucleic acid detection)	1.6 aM	KRAS & EGFR mutation	PCR or tHDA + Ago-based cleavage	- Requirement of nucleic acid amplification step (Multiple steps) - Cannot be proceeded on isotherm	[S6]
MAIDEN (Mesophilic Ago-based isothermal detection method)	4 nM	SARS-CoV-2	Reverse transcription + Ago-based cleavage	- Requirement of reverse transcription step (Multiple steps) - Low sensitivity	[S7]
SPOT (Scalable and portable Testing)	0.44 copies/uL	SARS-CoV-2	RT-LAMP + Ago-based cleavage	- Requirement of reverse transcription and nucleic acid amplification steps (Multiple steps) - Cannot be proceeded on isotherm	[S8]

Method	LOD	Target	Mechanism	Property	Ref
TtAgoEAR (TtAgo-based thermostable exponential amplification reaction)	10 aM	SARS-CoV-2	Ago-based cleavage + EXPAR	- Requirement of lots of proteins	[S9]
NOTE-Ago (Novel and One-step cleavage method based on Argonaute by integrating Tag-specific primer extension and Exonuclease I)	1 CFU/mL	S. Typhi S. aureus	PCR + Exo I-based digestion + Ago-based cleavage	- Requirement of nucleic acid amplification step (Multiple steps) - Cannot be proceeded on isotherm	[S10]
ANCA	1.87 fM (KPC) 178 aM (IMP) 529 aM (VIM) 120 aM (NDM) 144 aM (OXA-48)	CPKP	Ago-based cleavage with nucleic acid circuit	- One-step - Amplification-free - Isothermal	This work

Supplementary Table 3 Comparison of ANCA method with previous CPE detection methods.

Method	Sensitivity	LOD	Advantage	Limitation	Ref
MHT (Culture-based Modified Hodge Test)	>69%		- Simple and cost-effective	- Requires pure culture - Long reaction time for culturing - False-positive and false-negative	[S11]
CIM (Carbapenem-inactivation methods)	>90%		- Cover all carbapenemase - Simple and cost-effective	- Requires pure culture - Long reaction time for culturing	[S12]
Colorimetric assay based CarbaNP test and its automated kits	>70%		- Cover all carbapenemase - Simple and cost-effective - Low false-positive rate than MHT	- Hard to detect OXA-48 type producer - Long reaction time for culturing	[S13]
MALDI-TOF	>92%		- Accurate data analysis - High-throughput	- Requires expertise to analyze data - Requirement expensive equipment - Long reaction time for culturing	[S14]
LAMP	>90%		- Rapid and moderate cost - Applicable in limited-resource settings	- Contamination due to amplification of target nucleic acid - Complex primer design - Requirement of DNA extraction	[S15]
Direct detection of rectal swabs by a multiplex lateral flow immunoassay	80-100%		- Simple and cost-effective - Applicable in limited-resource settings	- Poor sensitivity - Long reaction time for culturing	[S16]
SERS combined with chemometric tool	99.8%		- Rapid and moderate cost - High sensitivity	- Poor reproducibility - Requires specific instrument and multivariate data analysis	[S17]
DNA microarray	99.4%	10 ³ copies/μL (~1.66 fM)	- High multiplexing - High sensitivity	- Requirement of expensive equipment and DNA extraction - Long reaction time for culturing	[S18]
RPA + CRISPR/Cas12	99.29%	4.48 fM	- High sensitivity and specificity - Excellent accuracy	- Contamination due to amplification of target nucleic acid - Multiple reaction steps	[S19]

Method	Sensitivity	LOD	Advantage	Limitation	Ref
LAMP + CRISPR/Cas12 + microfluidic chip		30 copies/reaction (~ 0.88 aM)	 - On-site detection - Naked eye detection - No complicated device 	 - Multiple reaction steps - Requirement of DNA extraction 	[S20]
CRISPR/Cas12 + SWV		3.5 fM	 - Excellent selectivity - Simple and cost-effective 	 - Requirement of DNA extraction - Multiple reaction steps 	[S21]
ECL		0.05 copies/ μ L (~0.08 aM)	 - High sensitivity - Amplification-free 	 - Requirement of DNA extraction - Multiple and long reaction steps 	[S22]
ANCA	100%	1.87 fM (KPC) 178 aM (IMP) 529 aM (VIM) 120 aM (NDM) 144 aM (OXA-48)	 - One-step - Amplification-free - Isothermal 	 - Ago-based cleavage with nucleic acid circuit 	This work

(Question 3) Line 386-387. Why choose “micropig skin, desk, glove, and scissors” as research subjects here? Random? Is there any special standard?

Response) The selection of micropig skin was motivated by its anatomical and physiological similarities to human skin, making it a widely accepted substitute for human skin in research. Due to the challenges of using human skin in such experimental settings, micropig skin serves as a stable and reliable alternative.

For the remaining items (desk, glove, and scissors), our selection was based on previous studies that included these surfaces. However, as you correctly noted, the underlying rationale was not explicitly discussed in these papers. To address this, we consulted the Standard Precautions for Healthcare Associated Infections and found guidelines suggesting that medical device surfaces such as handles and tweezers should be targeted for frequent disinfection.

With this in mind, we felt it would be beneficial to extend our experiments to other relevant surfaces. We therefore conducted additional experiments on doorknob and tweezers to further validate the robustness of our ANCA method in a clinical context. These new results have been incorporated into Figure 6 and discussed in the manuscript.

We believe this additional work effectively addresses your question and adds depth to our research, making it more clinically relevant. Again, we sincerely thank you for your insightful comments, which helped to improve the manuscript.

[Results]

Capture and detection of antibiotic-resistant bacteria using ANCA method with 3D nanopillar swab

To efficiently capture bacteria from surfaces, a 3D nanopillar array swab was employed. We first selected micropig skin as a substitute for human skin, which is reported to have properties very similar to human skin, to create an environment where direct contact occurs. We also selected a desk, glove, scissors, knob, and tweezers as experimental materials to create an environment where indirect contact occurs. These objects are mentioned in the Korean Standard Prevention Guidelines for Healthcare Associated Infections as requiring frequent disinfection. For the experiments, KPC- and IMP-producing bacteria were intentionally sprayed on micropig skin, desk, glove, scissors, knob, and tweezers. The swabs were then used

to capture bacteria on these surfaces by gently touching and rubbing them. Subsequently, the bacteria-captured swab was placed directly into a tube and the ANCA reaction was carried out. The complex 3D nanopillar array structure facilitates bacterial adhesion through nanotopographical interactions, allowing irreversible capture of bacteria and thus preventing secondary infection⁵². Fig. 6b shows the scanning electron microscope (SEM) images of bare and CPKP-captured complex nanopillar array structures. Bacteria were successfully captured on the swab. Fig. 6c and d show the results of KPC and IMP detection using the ANCA method for each target. For the KPC circuit, fluorescence signals were rapidly amplified only when *K. pneu* (KPC) was captured by the 3D nanopillar swab. Similarly, for the IMP circuit, signals increased only when *K. pneu* (IMP) was captured. These results suggest that on-site capture and identification of CPKP are feasible using the ANCA method in combination with the 3D complex nanopillar array swab.

[Figure]

Fig. 6 Capture and detection of antibiotic-resistant bacteria using ANCA method with 3D nanopillar swab. (a) Photograph of 3D nanopillar array swabs for the capture of CPKP from micropig skin, desk, scissors, glove, knob, and tweezers. Each bacterium was intentionally sprayed on the surfaces. CPKP-captured swab was directly used for ANCA reaction. (b) SEM images of 3D nanopillar array swabs before (upper) and after (lower) the capture of CPKP. (c, d) Plot of $T_0 - T$ value as a function of contaminated surface using (c) KPC and (d) IMP circuits

($n = 3$, error bar = standard deviation). T_0 represents the threshold time for negative control sample (absence of bacteria), and T denotes the threshold time for test sample (presence of bacteria).

(Question 4) Please describe the disadvantages of ANCA method used in this paper in the discussion section.

Response) In the Discussion section, we have taken the opportunity to address some of the limitations of the ANCA method. Although our technique offers several advantages, such as simplicity, rapidity, and the potential for high specificity, it is not without drawbacks. By acknowledging these limitations, we hope to guide future research aimed at refining the ANCA method and expanding its range of applications. We thank the reviewer for encouraging a balanced discussion of this topic.

[Discussion]

The ANCA method represents a pioneering attempt to synergize artificial nucleic acid circuits with Ago proteins. Consequently, its relatively low LOD can be attributed to the early stage of experimental optimization, especially when compared with established modalities for the detection of pathogenic microorganisms. Future avenues of research may explore enhancing the cleavage efficiency of the Ago protein/guide DNA through an iterative accumulation of experimental knowledge. Nevertheless, an evaluation of the C_t values associated with the clinical samples studied in this investigation reveals parallels with C_t values documented in the contemporary clinical literature⁶⁴⁻⁶⁶. Such congruence suggests that the ANCA technique has adequate robustness for clinical applications. The relatively long reaction time compared to other technologies is a drawback of ANCA. This is inevitable to inhibit the apo-form of the Ago protein. The apo-form is often formed in the absence of guide DNA, causing a non-specific cleavage reaction of dsDNA. To inhibit this, the concentration of Ago protein should not be too high. However, due to the nature of the ANCA technique, the higher the concentration of Ago protein, the more favorable the reaction kinetics. Therefore, it is necessary to choose a concentration condition that moderately addresses both issues, which led us to choose 200 nM Ago protein. However, we are hopeful that this issue will be resolved in the near future. This expectation is based on recent advances in protein engineering, such as inactivated Cas proteins. As research into the function of each domain of the Cas9 nuclease continues, Cas9 nucleases that cut only one strand or inactivated Cas9 nucleases with inhibited cleavage activity have

emerged. Given the increased interest in Ago protein research, we anticipate that an Ago protein with an inhibited apo-form will be available. Even in the absence of such protein engineering, the reaction rate can be increased by incorporating additional amplification-type circuitry, such as entropy-driven circuitry or catalytic hairpin assembly. Another drawback is that the data presented in the current NEB guide to DNA design is highly variable. This would explain the difference in sensitivity values between each circuit of the ANCA technology. As more data is collected, it may be possible to detect different target nucleic acids with consistent efficiency.

Reply to Reviewer 3

(Comment) Hyowon Jang *et al.* reported a single-enzyme, amplification-free, and isothermal detection of nucleic acids by combing artificial nucleic acid circuit with Argonaute (Ago) protein (ANCA). Using the ANCA assay, they detected the Carbapenemase-producing Enterobacteriaceae (CPE), including *Klebsiella pneumoniae* carbapenemase (KPC) and Imipenemase (IMP). Especially, they demonstrated that the ANCA assay could be directly employed to detect nucleic acids in human blood and urine, without need for nucleic acids extraction and amplification steps. Further, they validated the clinical utility of the ANCA assay by detecting both KPC and IMP in clinical samples, achieving a comparable performance with conventional PCR method. Although different catalytic nucleic acid circuits (including CRISPR-based catalytic nucleic acid circuit) have been designed and constructed for nucleic acid detection in previous literature, it is potentially novel and interesting to utilize Ago proteins to develop catalytic nucleic acid circuits for signal amplification detection of nucleic acids. However, I do have some concerns and comments below.

Response) Thank you for your thorough review and positive comments on the potential novelty and interest of our ANCA method. We are encouraged by your recognition of the clinical utility of our assay and its performance compared to conventional PCR methods. We acknowledge your concerns and have taken steps to address them in the revised manuscript.

(Question 1) Figure 1 needs to be re-designed because it is similar to Figure 2a. In addition, it does not clearly explain the working principle of the ANCA assay. For instance, in the main manuscript, the author mentioned that the 5' end of the guide DNA plays an important role in binding to the Ago protein. However, there is no indication of their 5' end and 3' end in Figure 1.

Response) Thank you for pointing out the shortcomings of Figure 1. We fully agree that the figure needs to be clearer to better explain the principle of the ANCA assay. Following your suggestions, we have redesigned Figure 1 to avoid redundancy with Figure 2a and to include the missing details.

In particular, we have labeled the 5' end of the guide DNA with a "P" to emphasize that it is modified with a phosphate, which is critical for binding to the Ago protein.

In addition, to make the steps of the assay easier to understand, we have included new supplementary figures showing the 5' and 3' ends of each sequence involved in the ANCA technique. These new figures are intended to provide the reader with an intuitive understanding of which sequences are hybridized and cleaved. We have added a total of five such supplementary figures, one for each target nucleic acid studied.

We believe these changes will help clarify the mechanism of ANCA and thank you again for your insightful comments.

[Figure]

Fig. 1 Schematic illustration of ANCA method for the detection of target nucleic acid. First, Ago/G1 and Ago/G2 complexes hybridize and cleave target DNA, generating T1. Second, Ago/T1 complex forms, and it hybridizes and cleaves R, releasing Output and T2. Third, Ago/T2 complex forms, and it hybridizes and cleaves R*, generating T1 and thus completing ANCA. In the presence of target DNA, ANCA is repeatedly carried out and fluorescence signal increases. P stands for phosphate attached to the 5' end.

[Supplementary Information]

Supplementary Fig. 1 Sequence-based illustration of KPC circuit.

Supplementary Fig. 2 Sequence-based illustration of IMP circuit.

Supplementary Fig. 3 Sequence-based illustration of VIM circuit.

Supplementary Fig. 4 Sequence-based illustration of NDM circuit.

Supplementary Fig. 5 Sequence-based illustration of OXA-48 circuit.

(Question 2) There are several concerns in Figure 2: i) even if ANCA assay is referred to as a positive feedback circuit, it seems that fluorescence signals should be observed even with only R present. Why is there no signal detected at all (Fig. 2d)? ii) more control groups are needed to evaluate the ANCA method. For example, there is no result to show the cleavage of the target sequence by the Ago/G1 and Ago/G2 or either one. It is unclear whether the T1 can be produced or released by the cleavage reaction between Ago and the target. iii) The experimental explanation in the manuscript and the figure are very vague and unclear. Does Ago protein mean Ago/G1 and Ago/G2 in Figure 2c and d?

Response) We appreciate the reviewer's detailed feedback, which has given us the opportunity to improve the clarity and robustness of our manuscript. We have addressed each of their concerns and made the following changes:

- i) Regarding the lack of fluorescence signal in Figure 2d when only R is present, it's important to note that the cleavage rate in this condition is too slow to amplify the fluorescence signal to a detectable level. We have now repeated the experiment with a 10-fold increase in the concentration of target DNA, which allowed us to observe a slow amplification of the fluorescence signal over time when only R is present. The results are shown in the supplementary material. We also confirmed the cleavage of R by performing PAGE analysis, which showed a band corresponding to degraded R. This new data is also included and discussed in the text.

- ii) We acknowledge the need for more control groups in Figure 2 to fully evaluate the ANCA method. As suggested, we have re-ran the electrophoresis analysis and added more control groups to demonstrate how the bands form under different conditions, both in the presence and absence of all components. This should clarify the cleavage capabilities of Ago/G1 and Ago/G2 and their interactions with the target.
- iii) We apologize for the vagueness and lack of clarity in the explanations provided in the manuscript and in Figure 2. We have reviewed the entire manuscript to remove any ambiguities and to standardize the terminology used.

We hope that these revisions effectively address the reviewer's concerns, and we are grateful for the suggestions that have led us to improve our work.

[Results]

Evaluation of ANCA method

The ANCA method was designed to provide exponential amplification through a positive feedback loop (Fig. 2a). We first evaluated the ANCA method with different components using KPC as the target. In the presence of all reaction components, including the target, strong fluorescence signals were observed during the ANCA reaction (spectrum 5 in Fig. 2c). In the absence of target DNA, Ago protein, G1, or G2, negligible signals were detected (spectra 1-4 in Fig. 2c). The feasibility of the ANCA method is further confirmed by the results of polyacrylamide gel electrophoresis (PAGE) analysis (right panel in Fig. 2c). The band indicating the intact R-R* structure (lane M2 in Fig. 2c) was completely degraded after the ANCA reaction (lane 5 in Fig. 2c). On the other hand, in the absence of target DNA, Ago protein, G1, or G2, the bands corresponding to the R-R* structure remain intact (lanes 1-4 in Fig. 2c). To evaluate the amplification efficiency of the ANCA method, we performed comparative experiments (Fig. 2b). In the ANCA reaction without R*, the Ago/G1 and Ago/G2 complexes cleave the target DNA, resulting in the formation of the Ago/T1 complex. The Ago/T1 complex then cleaves R, releasing Output and T2. However, no further reaction occurs because of the absence of R*. As a result, no fluorescence signals were observed even after 400 min of reaction, in the presence of the target sequence (spectrum 5 in Fig. 2d). In addition, no signals were detected in the absence of target DNA, Ago protein, G1, or G2 (spectra 1-4 in Fig. 2d). The PAGE results also show the intact R structure bands for all conditions (right panel in Fig. 2d).

To verify that the higher concentration of target DNA could be detected with R alone, we repeated the experiment using a 10-fold increase in the concentration of target DNA used in Fig. 2. As a result, we could see that the fluorescence signal increased over time in the presence of R alone, although at a slower rate than in the presence of R and R* together (spectrum 5 in Supplementary Fig. 6). The PAGE results showed a fading of the band corresponding to R, where all components were present, and the production of a shorter band, which we speculate is due to the degradation of R (lane 5 in Supplementary Fig. 6). This evidence indicates that the ANCA method is a positive feedback circuit that provides high amplification efficiency and facilitates sensitive detection of target DNA.

[Figure]

Fig. 2 Evaluation of ANCA method. (a, b) Schematic illustration of (a) ANCA and (b) ANCA without R*. ANCA method enables exponential amplification through positive feedback. ANCA without R* cleaves target DNA and R, but no further reaction transpires. (c) Time-dependent fluorescence intensities during ANCA reaction (left) and corresponding PAGE analysis result (right) under various components. (d) Time-dependent fluorescence intensities during ANCA reaction without R* (left) and corresponding PAGE analysis result (right) under

various components. [Target DNA] = 1 nM, [R] = 500 nM, [R*] = 500 nM, [G1] = 25 nM, [G2] = 25 nM, [Ago] = 200 nM, [MgCl₂] = 10 mM, [NaCl] = 75 mM, and [BSA] = 10 μg/mL.

[Supplementary Information]

Supplementary Fig. 6 Time-dependent fluorescence intensities during ANCA reaction without R* (left) and corresponding PAGE analysis result (right) under various components. [Target DNA] = 10 nM, [R] = 500 nM or 1000 nM (only MI), [G1] = 25 nM, [G2] = 25 nM, [Ago] = 200 nM, [MgCl₂] = 10 mM, [NaCl] = 75 mM, and [BSA] = 10 μg/mL.

(Question 3) In Figure 4, the authors demonstrated that the bacteria in raw samples (e.g., urine, blood) can be directly detected without DNA purification, how did they address the issue of autofluorescence (or background) caused by raw samples themselves (e.g., especially for blood in Figure 4g)?

Response) We thank the reviewer for raising the important issue of autofluorescence in raw samples, especially in blood, as shown in Figure 4g. We agree that autofluorescence could potentially confound the results.

Prior to performing this experiment, we also had concerns about autofluorescence, but found that it did not significantly affect our detection performance. To address this issue, we performed a comparison of autofluorescence by examining the raw fluorescence data without background subtraction. We looked specifically at the fluorescence signal in the FAM wavelength region and found negligible differences between the raw sample and distilled water.

We believe that the high concentration of reporter used in our experiments mitigated the effect of any background signal from the FAM dye attached to the reporter. To make this clear, we have included the data comparing autofluorescence between raw samples and distilled water in the supplementary material and have elaborated on this point further in the manuscript.

We hope this addresses any concerns about the potential for autofluorescence to confound our results.

[Results]

Direct detection of antibiotic-resistant bacteria using ANCA method

In addition, we performed tests on urine and blood samples to determine the robustness of the experiment against autofluorescence. When evaluating the fluorescence signal within the FAM wavelength range, we observed minimal differentiation between the samples and distilled water (Supplementary Fig. 14).

[Supplementary Information]

Supplementary Fig. 14 (a, b) Time-dependent fluorescence intensities from ANCA mixture with (a) urine and (b) blood samples.

(Question 4) They stated that the 3D nanopillar array structure enables the irreversible entrapment of bacteria. In the method section, it was mentioned that the bacteria-captured nanopillar structure was immersed in distilled water. However, it does not make sense that irreversibly entrapped bacteria can be easily separated from the nano-structure by simply soaking. How can the captured bacteria be released for analysis?

Response) We apologize for the oversight and thank the reviewer for pointing out this inconsistency in the Methods section of our manuscript.

You are correct that the 3D nanopillar array structure is designed for irreversible entrapment of bacteria. After the bacteria are captured using the 3D nanopillar array, the entire structure is immersed in the ANCA reaction solution to perform the subsequent steps. The ANCA reaction takes place at high temperatures, allowing the captured bacterial cells to be degraded and their DNA released into the solution. Thus, even if the bacteria are not physically separated from the 3D structure, the assay can still proceed effectively due to this cell lysis and DNA release. Previous literature suggests that bacteria captured on the 3D nanopillar array structure can be lysed and the released nucleic acid can be detected (*Nano Conver.* **2023**, *10*, 25.; *ACS Nano* **2021**, *15*, 4777.; *ACS Nano* **2020**, *14*, 17241.)

We have revised this section in the manuscript to clarify the procedure and eliminate any confusion. Thank you again for your keen observation, which helped to improve the quality of our work.

[Experimental Section]

Capture and detection of antibiotic-resistant bacteria using ANCA method with 3D nanopillar swab. The 3D nanopillar array swabs were prepared as described previously^{50, 61}. Each bacterium (10^5 cells/mL) was dropped onto a micropig Franz cell membrane (APURES, Seoul, Korea), desk, scissors, glove, knob, and tweezers. Bacteria were collected from the contaminated surfaces using 3D nanopillar swabs by simple touching and rubbing. SEM images were taken using a Nova 230 system (FEI, Hillsboro, OR, USA) at an accelerating voltage of 15 keV. The piece of bacteria-captured 3D nanopillar swab was mixed with the ANCA reaction components (18 μ L), and distilled water (2 μ L) was added to make the final volume of 20 μ L. The entire mixture was then incubated following the aforementioned protocol.

(Question 5) In the ANCA method, its limit of detection (LoD) of fM (18.2 fM for KPC and 2.5 fM for IMP) are not very impressive. Its LOD may be lower when testing clinical samples. Typically, PCR method can achieve a LoD of aM when testing extracted or purified nucleic acid. However, in their clinical validation, it seems that the extraction-free ANCA method could achieve a comparable performance with PCR method (Figure 5). Did the authors extract DNA before running PCR testing? If so, how could their extraction-free, amplification-free ANCA method achieve a sensitivity of 100% and specificity of 100% compared with PCR method (including nucleic acid extraction and enzymatic amplification)?

Response) We appreciate the reviewer's astute observation regarding the sensitivity of our ANCA method. While it's true that our originally reported LOD values were relatively low compared to other conventional methods for the detection of pathogenic microorganisms, we took corrective action after submission.

During further investigation, we identified an influential component within the enzyme storage buffer, distinct from the reaction buffer, that was affecting performance. By optimizing the concentration of this component (Supplementary Figure 8), along with other critical components, we were able to significantly improve the sensitivity of our method. The resulting LOD values for the KPC and IMP circuits were 1.87 fM and 178 aM, respectively, as shown in revised Figure 3.

In addition, we have successfully extended our work to the detection of VIM, NDM and OXA-48, thereby enhancing the clinical utility of the ANCA technology (Supplementary Figure 9 and 12). The inclusion of these genes, along with KPC and IMP, is critical because these five genes are integral to CPE as they are most often responsible for conferring resistance to carbapenems.

It's important to note that the revised sensitivity levels are well aligned with the clinical context. As shown in Figure 5, the ANCA method has demonstrated its ability to detect clinically relevant specimens, which showed C_t values ranging from 12 to 33, mirror the C_t values reported in established medical research. These parallels underscore the ability of ANCA to effectively identify typical clinical specimens.

In light of these findings, we believe that ANCA sensitivity remains relevant in the clinical setting. In essence, the ability to detect authentic KPCs and IMPs does not require excessive sensitivity. An overemphasis on ultra-high sensitivity could inadvertently increase false-positive rates, leading to unwarranted clinical concerns.

In response to the reviewer's question, we did use extracted DNA for our PCR procedures. However, we suspect that the ANCA technique was able to produce results comparable to PCR because of nucleic acid losses during the extraction phase. While the nucleic acid extraction method provides purified nucleic acids, there is an unavoidable loss due to the repeated lysis

and washing steps. Conversely, when using direct rectal swabs, experiments can be performed without this nucleic acid attrition, leading to the observed results. We have clarified the specifics of the nucleic acid extraction procedure in the Experimental Section and included this clarification in the revised manuscript.

We have created an additional comparison table that compares the sensitivity of the ANCA method to other established techniques. Interestingly, the sensitivity of ANCA was found to be superior when compared to other Ago protein-based detection methods without pre-amplification. This finding highlights the potential scalability and improvement opportunities for ANCA within the field of Ago protein-based techniques.

We have included the aforementioned performance comparison table in the Supplementary Information and made the necessary changes to the manuscript to reflect these clarifications. Thank you again for your valuable feedback, which has helped us to improve the quality of our work.

[Results]

Optimization and sensitivity of ANCA method

To determine the optimal conditions for the ANCA method, we first monitored changes in fluorescence by varying R and R* (Supplementary Fig. 7). When the fluorophore and quencher were attached to the R sequence only, the ANCA method demonstrated efficient target DNA detection. It is worth noting that when the fluorophore and quencher were attached to both R and R*, the reaction efficiency was significantly reduced, which may be attributed to steric hindrance affecting the cleavage activity of Ago/guide DNA complexes. We then optimized the reaction temperature and the concentrations of R, R*, Ago protein, and Mg²⁺ for the ANCA method. As shown in Supplementary Fig. 8a-e, the conditions for optimal detection performance were found to be a reaction temperature of 75 °C and concentrations of 500 nM for R and R*, 200 nM for Ago protein, and 10 mM for MgCl₂. In addition, the concentrations of NaCl and bovine serum albumin (BSA) in the enzyme storage buffer were optimized to 75 mM and 10 µg/mL, respectively (Supplementary Fig. 8f and g). These conditions were used in subsequent experiments.

Under the optimized conditions, we evaluated the sensitivity of the ANCA method for KPC and IMP as target sequences, respectively. Fig. 3a shows the time-dependent fluorescence intensities for different concentrations of the KPC sequence. The threshold time was determined as the reaction time at which the fluorescence intensity reached 10,000. Fig. 3b shows the plot of threshold time as a function of target DNA concentration. A linear relationship ($R^2 = 0.99$) was observed in the range of 10 fM to 10 nM, and the limit of detection (LOD) was calculated to be 1.87 fM using the formula³⁹ (limit of blank (LOB) = mean of blank + 1.645 × standard deviation of blank, LOD = LOB + 1.645 × standard deviation of low concentration sample). Additionally, we adapted the ANCA method for the detection of IMP. Using the IMP circuit, a linear relationship ($R^2 = 0.99$) was observed in the range of 1 fM to 1 nM, and the LOD was calculated to be 178 aM (Fig. 3c and d). Similarly, circuits for the detection of VIM, NDM, and OXA-48 were fabricated and evaluated for their sensitivity to each of the target substances. In all cases, a linear relationship ($R^2 = 0.99$) was observed in the range of 1 fM to 1 nM, and the LOD of VIM, NDM, and OXA-48 were calculated to be 529, 120, and 144 aM, respectively (Supplementary Fig. 9). The sensitivity of the ANCA method is far superior to previous results that utilize Ago proteins without pre-amplification (Supplementary Table 2)²⁷⁻³⁶. Furthermore, it is noteworthy that the ANCA method demonstrates flexibility and broad target applicability, as evidenced by the successful detection of KPC, IMP, VIM, NDM, and OXA-48 sequences by slight modifications of the ANCA method.

Direct detection of antibiotic-resistant bacteria using ANCA method

We also attempted to detect CPKP using the VIM, NDM, and OXA-48 circuits and found that each circuit was able to discriminate well between WT and CPKP. (Supplementary Fig. 12a-c).

[Discussion]

Unlike PCR, the ANCA method performs all reactions isothermally without purification, simplifying the procedure and reducing the need for extensive equipment. During the repeated lysis and washing process for nucleic acid extraction, some nucleic acids are inevitably lost. In the ANCA method, rectal swabs and transport media are used directly, allowing the detection of DNA without loss.

The ANCA method represents a pioneering attempt to synergize artificial nucleic acid circuits with Ago proteins. Consequently, its relatively low LOD can be attributed to the early stage of

experimental optimization, especially when compared with established modalities for the detection of pathogenic microorganisms. Future avenues of research may explore enhancing the cleavage efficiency of the Ago protein/guide DNA through an iterative accumulation of experimental knowledge. Nevertheless, an evaluation of the C_t values associated with the clinical samples studied in this investigation reveals parallels with C_t values documented in the contemporary clinical literature⁶⁴⁻⁶⁶. Such congruence suggests that the ANCA technique has adequate robustness for clinical applications.

[Figure]

Fig. 3 Sensitivity of ANCA method. (a, c) Time-dependent fluorescence intensities during ANCA method with various concentrations of (a) KPC and (c) IMP sequences. Dashed black threshold lines indicate the reaction time at which the fluorescence intensity reached 10,000 (threshold time). (b, d) Correlation of threshold time to the logarithm of (b) KPC and (d) IMP concentration ($n = 3$, error bar = standard deviation). Red lines are linear fits, indicating LODs of 1.87 fM for KPC and 178 aM for IMP. $[R] = 500$ nM, $[R^*] = 500$ nM, $[G1] = 25$ nM, $[G2] = 25$ nM, $[Ago] = 200$ nM, $[MgCl_2] = 10$ mM, $[NaCl] = 75$ mM, and $[BSA] = 10$ μ g/mL.

[Experimental Section]

For qRT-PCR, DNA was extracted from antibiotic-resistant bacteria using the Monarch® Plasmid Miniprep Kit. Extracted DNA (2 µL) was combined with a PCR reaction mixture (18 µL) containing the gene-specific primer set (0.5 µM each), 1× TOPsimple preMIX (aliquot)-nTaq, and 1× SYBR Green I dye. Amplification was then proceeded with the following steps: 49 cycles of denaturation for 30 s at 95 °C, annealing for 1 min at 55 °C, and extension for 1 min at 72 °C. The fluorescence signal from the PCR was monitored at each extension step using a CFX Opus 96 RT-PCR system.

Prior to applying the ANCA method, qRT-PCR was performed to confirm the presence of CPKP-induced gene in the bacteria-spiked human sample. DNA was extracted using the Monarch® Plasmid Miniprep Kit and PCR was performed following the aforementioned protocol.

For the comparison of ANCA method with qRT-PCR, DNA was extracted from the rectal swab-immersed transport media using the AccuPrep® Stool DNA Extraction Kit. The extracted DNA (2 µL) was combined with PCR reaction mixture (18 µL) containing the KPC gene-specific primer set (0.5 µM each), 1× TOPsimple preMIX (aliquot)-nTaq, and 1× SYBR Green I dye. The fluorescence signal at each extension step was recorded using the CFX Opus Real-Time System (Bio-Rad). The accompanying software (CFX Maestro) was used to obtain C_t values.

[Supplementary Information]

Supplementary Fig. 8 (a-g) Plot of $T_0 - T$ value by varying (a) reaction temperature and concentrations of (b) R, (c) R*, (d) Ago protein, (e) MgCl₂, (f) NaCl, and (g) BSA for ANCA method (n = 3, error bar = standard deviation, N.D. = threshold time was not determined). When the reaction temperature was 75 °C and the concentrations of R, R*, Ago protein, MgCl₂, NaCl, and BSA were 500 nM, 500 nM, 200 nM, 10 mM, 75 mM, and 10 µg/mL, respectively, the highest $T_0 - T$ values were obtained. T_0 represents the threshold time for negative sample, and T denotes the threshold time for positive sample. [Target DNA] = 10 nM.

Supplementary Fig. 9 (a, c, e) Time-dependent fluorescence intensities during ANCA method with various concentrations of (a) VIM, (c) NDM, and (e) OXA-48 sequences. Dashed black threshold lines indicate the reaction time at which the fluorescence intensity reached 10,000 (threshold time). (b, d, f) Correlation of threshold time to the logarithm of (b) VIM, (d) NDM, and (f) OXA-48 concentration ($n = 3$, error bar = standard deviation). Red lines are linear fits, indicating LODs of 529 aM for VIM, 120 aM for NDM, and 144 aM for OXA-48. $[R] = 500$ nM, $[R^*] = 500$ nM, $[G1] = 25$ nM, $[G2] = 25$ nM, $[Ago] = 200$ nM, $[MgCl_2] = 10$ mM, $[NaCl] = 75$ mM, and $[BSA] = 10$ μ g/mL.

Supplementary Fig. 12 (a-c) Plot of $T_0 - T$ value as a function of bacteria in buffer using (a) VIM, (b) NDM, and (c) OXA-48 circuits ($n = 3$, error bar = standard deviation). T_0 represents the threshold time for negative control sample (absence of bacteria), and T denotes the threshold time for test sample (presence of bacteria).

Supplementary Table 2 Comparison of ANCA method with previous Ago-based nucleic acid detection methods.

Method	LOD	Target	Mechanism	Property	Ref
Ago-FISH (Argonaute-based fluorescence in situ hybridization)	1 pM	miRNA (Let7a, c)	Ago-based cleavage	- Low sensitivity	[S1]
MULAN (Multiplex Argonaute-based nucleic acid detection system)	1.6 copies/reaction	SARS-CoV-2 Influenza virus	RT-LAMP + Ago-based cleavage	- Requirement of reverse transcription and nucleic acid amplification steps (Multiple steps) - Cannot be proceeded on isotherm	[S2]
NAVIGATER (Nucleic acid enrichment via DNA guided Argonaute from thermus thermophilus)	1 copy/reaction	KRAS mutation	PCR or RT-LAMP + Ago-based cleavage	- Requirement of nucleic acid amplification step (Multiple steps) - Cannot be proceeded on isotherm	[S3]
A-Star (Ago-directed specific target enrichment and detection)	1 copy/reaction	BRAF & EGFR mutation	PCR + Ago-based cleavage	- Requirement of nucleic acid amplification step (Multiple steps) - Cannot be proceeded on isotherm	[S4]
PLCR (Pfago coupled with modified ligase chain reaction for nucleic acid detection)	10 aM	SARS-CoV-2	Ligase chain reaction + Ago-based cleavage	- Requirement of reverse transcription and nucleic acid amplification steps (Multiple steps) - Cannot be proceeded on isotherm	[S5]
PAND (Pfago-mediated nucleic acid detection)	1.6 aM	KRAS & EGFR mutation	PCR or tHDA + Ago-based cleavage	- Requirement of nucleic acid amplification step (Multiple steps) - Cannot be proceeded on isotherm	[S6]
MAIDEN (Mesophilic Ago-based isothermal detection method)	4 nM	SARS-CoV-2	Reverse transcription + Ago-based cleavage	- Requirement of reverse transcription step (Multiple steps) - Low sensitivity	[S7]
SPOT (Scalable and portable Testing)	0.44 copies/uL	SARS-CoV-2	RT-LAMP + Ago-based cleavage	- Requirement of reverse transcription and nucleic acid amplification steps (Multiple steps) - Cannot be proceeded on isotherm	[S8]

Method	LOD	Target	Mechanism	Property	Ref
TtAgoEAR (TtAgo-based thermostable exponential amplification reaction)	10 aM	SARS-CoV-2	Ago-based cleavage + EXPAR	- Requirement of lots of proteins	[S9]
NOTE-Ago (Novel and One-step cleavage method based on Argonaute by integrating Tag-specific primer extension and Exonuclease I)	1 CFU/mL	S. Typhi S. aureus	PCR + Exo I-based digestion + Ago-based cleavage	- Requirement of nucleic acid amplification step (Multiple steps) - Cannot be proceeded on isotherm	[S10]
ANCA	1.87 fM (KPC) 178 aM (IMP) 529 aM (VIM) 120 aM (NDM) 144 aM (OXA-48)	CPKP	Ago-based cleavage with nucleic acid circuit	- One-step - Amplification-free - Isothermal	This work

Supplementary Table 3 Comparison of ANCA method with previous CPE detection methods.

Method	Sensitivity	LOD	Advantage	Limitation	Ref
MHT (Culture-based Modified Hodge Test)	>69%		- Simple and cost-effective	- Requires pure culture - Long reaction time for culturing - False-positive and false-negative	[S11]
CIM (Carbapenem-inactivation methods)	>90%		- Cover all carbapenemase - Simple and cost-effective	- Requires pure culture - Long reaction time for culturing	[S12]
Colorimetric assay based CarbaNP test and its automated kits	>70%		- Cover all carbapenemase - Simple and cost-effective - Low false-positive rate than MHT	- Hard to detect OXA-48 type producer - Long reaction time for culturing	[S13]
MALDI-TOF	>92%		- Accurate data analysis - High-throughput	- Requires expertise to analyze data - Requirement expensive equipment - Long reaction time for culturing	[S14]
LAMP	>90%		- Rapid and moderate cost - Applicable in limited-resource settings	- Contamination due to amplification of target nucleic acid - Complex primer design - Requirement of DNA extraction	[S15]
Direct detection of rectal swabs by a multiplex lateral flow immunoassay	80-100%		- Simple and cost-effective - Applicable in limited-resource settings	- Poor sensitivity - Long reaction time for culturing	[S16]
SERS combined with chemometric tool	99.8%		- Rapid and moderate cost - High sensitivity	- Poor reproducibility - Requires specific instrument and multivariate data analysis	[S17]
DNA microarray	99.4%	10 ³ copies/μL (~1.66 fM)	- High multiplexing - High sensitivity	- Requirement of expensive equipment and DNA extraction - Long reaction time for culturing	[S18]
RPA + CRISPR/Cas12	99.29%	4.48 fM	- High sensitivity and specificity - Excellent accuracy	- Contamination due to amplification of target nucleic acid - Multiple reaction steps	[S19]

Method	Sensitivity	LOD	Advantage	Limitation	Ref
LAMP + CRISPR/Cas12 + microfluidic chip		30 copies/reaction (~ 0.88 aM)	 - On-site detection - Naked eye detection - No complicated device 	 - Multiple reaction steps - Requirement of DNA extraction 	[S20]
CRISPR/Cas12 + SWV		3.5 fM	 - Excellent selectivity - Simple and cost-effective 	 - Requirement of DNA extraction - Multiple reaction steps 	[S21]
ECL		0.05 copies/ μ L (~0.08 aM)	 - High sensitivity - Amplification-free 	 - Requirement of DNA extraction - Multiple and long reaction steps 	[S22]
ANCA	100%	1.87 fM (KPC) 178 aM (IMP) 529 aM (VIM) 120 aM (NDM) 144 aM (OXA-48)	 - One-step - Amplification-free - Isothermal 	 - Ago-based cleavage with nucleic acid circuit 	This work

(Question 6) The sequence of oligonucleotide T1 for KPC or IMP should be provided in the supplementary table 1. In addition, what is the reference gene used for PCR?

Response) We appreciate the reviewer's suggestion to include detailed information about the oligonucleotide sequences and reference genes. As recommended, we have added the sequence of T1 for both KPC and IMP detection in Supplementary Table 1 for completeness and ease of reference for the reader.

Regarding the reference gene for PCR, we used the ureR gene as an internal control. Primers for this gene were designed using the Primer 3 tool, as described in the Methods section of the manuscript. These primer sequences have also been included in the revised Supplementary Table 1 to provide a comprehensive set of information.

Thank you for your attention to detail, which helps to improve the quality and transparency of our work.

[Supplementary Information]

Supplementary Table 1 Sequences of oligonucleotide used in this experiment.

Name	Sequence (5' → 3')	Target
KPC guide DNA 1 (G1)	P ^(a) -TATCA CTGTA TTGCACG	KPC
KPC guide DNA 2 (G2)	P ^(a) -CAAAT TGGCG GCGGCGT	
KPC Reporter (R)	GCGGCGT TATCACTGTA ACTAAAT	
KPC Reporter (R)-fluorophore modified	FAM-GCGGCGT TATCACTGTA ACTAAAT-BHQ2	
KPC Reporter complement (R*)	ATTTAGT TACAGTGATA ACGCCGC	
KPC Reporter complement (R*)-fluorophore modified	BHQ2-ATTTAGT TACAGTGATA ACGCCGC-FAM	
KPC trigger (T1)	TACAGTGATA ACGCCGC	
KPC forward primer	GGTTCTGTGGTCACCCATCT	
KPC reverse primer	TCCAGACGGAACGTGGTATC	
KPC synthetic target DNA	GGTGGCGGAGCTGTCCGCGGCCCGTGCAA TACAGTGATA ACGCCG CCGCCAATTTGTTGC	
KPC synthetic target complementary DNA	GCAACAAATTGGCGGCGGCGTTTATCACTGTATTGCACGGCGGCCGCGGACAGCTCCGCCACC	
IMP guide DNA 1 (G1)	P ^(a) -TAACTTTTCA GTATCTT	IMP
IMP guide DNA 2 (G2)	P ^(a) -TCCACAAACC AAGTGAC	
IMP Reporter (R)	FAM-AAGTGAC TAACTTTTCA ACTAAAT-BHQ2	
IMP Reporter complement (R*)	ATTTAGT TGAAAAGTTA GTCACCTT	
IMP trigger (T1)	TGAAAAGTTA GTCACCTT	
IMP forward primer	CCTAAACATGGCTTGGTGGT	
IMP reverse primer	GCATACGTGGGGATAGATCG	
IMP synthetic target DNA	CTAATTGACACTCCATTTACGGCTAAAGATACTGAAAAGTTAGTCACTTGGTTTGTGGAGCGTG	

Name	Sequence (5' → 3')	Target
IMP synthetic target complementary DNA	CACGCTCCACAAACCAAGTGACTAACTTTTCAGTATCTTTAGCCGTAATGGAGTGTCAATTAG	
VIM guide DNA 1 (G1)	P ^(a) -TAAACTACTA ATA AACTT	VIM
VIM guide DNA 2 (G2)	P ^(a) -GCGGTCATGT AGACCAA	
VIM Reporter (R)	FAM-AGACCAA TAAACTACTA ACTAAAT-BHQ2	
VIM Reporter complement (R*)	ATTTAGT TAGTAGTTTA TTGGTCT	
VIM trigger (T1)	TAGTAGTTTA TTGGTCT	
VIM synthetic target DNA	ATGTTAAAAGTTAT TAGTAGTTTATTGGTCTACATGACCGCGTCTGTCATGGCTGTCGCAAGTC	
VIM synthetic target complementary DNA	GACTTGCGACAGCCATGACAGACGCGGTCATGTAGACCAATAAACTACTAATAACTTTTAAACAT	
NDM guide DNA 1 (G1)	P ^(a) -TAAGTCGCAA TCCCCGC	NDM
NDM guide DNA 2 (G2)	P ^(a) -TCGACAACGC ATTGGCA	
NDM Reporter (R)	FAM-ATTGGCA TAAGTCGCAA ACTAAAT-BHQ2	
NDM Reporter complement (R*)	ATTTAGT TTGCGACTTA TGCCAAT	
NDM trigger (T1)	TTGCGACTTA TGCCAAT	
NDM synthetic target DNA	GGTATGGACGCGCTGCATGCGGCGGGGATTGCGACTTATGCCAATGCGTTGTCGAACCAGCTTG	
NDM synthetic target complementary DNA	CAAGCTGGTTTCGACAACGCATTGGCATAAGTCGCAATCCCCGCCGCATGCAGCGCGTCCATACC	
OXA-48 guide DNA 1 (G1)	P ^(a) -TTTCTTAAAA AGCTGAT	OXA-48
OXA-48 guide DNA 2 (G2)	P ^(a) -ACTTATTGTG ATACAGC	
OXA-48 Reporter (R)	FAM-ATACAGC TTTCTTAAAA ACTAAAT-BHQ2	
OXA-48 Reporter complement (R*)	ATTTAGT TTTTAAGAAA GCTGTAT	
OXA-48 trigger (T1)	TTTAAAGAAA GCTGTAT	
OXA-48 synthetic target DNA	ATTCTGAATTTTCGGCCACGGAGCAAATCAGCTTTTTAAGAAAGCTGTATCACAATAAGT TACACG	

Name	Sequence (5' → 3')	Target
OXA-48 synthetic target complementary DNA	CGTGTAACCTTATTGTGATACAGCTTTCTTAAAAAGCTGATTGCTCCGTGGCCGAAATTCGAAT	
ureR forward primer ureR reverse primer	GGATATCTGACCAGTCGG GGGTTTTGCGTAATGATCTG	WT

^(a)P indicates the modification of phosphate group at 5' end.

(Question 7) The authors should explain how the false positive signals were generated in their ANCA method. Why do their false positive curves have still a sigmoidal shape which is typically caused due to exponential amplification?

Response) We appreciate the reviewer's keen observation on the issue of false positive signals in our ANCA method. As noted, our method did generate false positive curves with a sigmoidal shape, which can give the impression of exponential amplification. The reason for this is due to a phenomenon known as "chopping" associated with Ttago in its apo-form (i.e., Ttago existing without guide DNA). In the apo-form, Ttago can lead to non-specific dsDNA cleavage.

Normally, the concentration of guide DNA should be at least 10 times higher than that of Ttago to prevent cleavage. However, the ANCA assay design requires Ttago to be present in the apo-form to complete the feedback loop with the newly generated trigger. Over time, this leads to non-specific cleavage of some reporter molecules, generating a fluorescence signal that can resemble true positive results.

In experiments with actual target nucleic acid, the rapid formation of the trigger reduces the concentration of Ttago in the apo-form, resulting in rapid and specific amplification of the fluorescence signal. However, non-specific cleavage by the apo-form is slower, and this difference allows discrimination of true and false positives.

We are investigating proteins that may prevent this non-specific cleavage in the apo-form, which would make the technique even more selective. We have updated our manuscript to include this discussion to provide a clearer understanding of the observed phenomena.

[Discussion]

The relatively long reaction time compared to other technologies is a drawback of ANCA. This is inevitable to inhibit the apo-form of the Ago protein. The apo-form is often formed in the absence of guide DNA, causing a non-specific cleavage reaction of dsDNA. To inhibit this, the concentration of Ago protein should not be too high. However, due to the nature of the ANCA technique, the higher the concentration of Ago protein, the more favorable the reaction kinetics. Therefore, it is necessary to choose a concentration condition that moderately addresses both issues, which led us to choose 200 nM Ago protein. However, we are hopeful that this issue will be resolved in the near future. This expectation is based on recent advances in protein

engineering, such as inactivated Cas proteins. As research into the function of each domain of the Cas9 nuclease continues, Cas9 nucleases that cut only one strand or inactivated Cas9 nucleases with inhibited cleavage activity have emerged. Given the increased interest in Ago protein research, we anticipate that an Ago protein with an inhibited apo-form will be available.

(Question 8) In line 290-292, they mentioned that “A strong correlation was observed between the $T_0 - T$ and $C_{10} - C_t$ values for KPC-positive clinical samples (Fig. 5c). C_{10} was set to 50 and C_t denotes the cycle threshold of PCR for clinical sample.” T_0 represents the threshold time for negative control sample or threshold time for the pure transport media. However, why did they set C_{10} to 50, not the threshold time of the pure transport like T_0 . In addition, in supplementary Figure 3, the cycle threshold of the negative sample is less 50.

Response) We appreciate the reviewer's attention to detail and apologize for the oversight in our original manuscript. To clarify, the C_{10} value of 50 was set because it represents the cycle threshold when PCR is performed on pure transport media, similar to how T_0 represents the threshold time for the negative control sample or pure transport media. We understand that the language used was not clear and we have made the necessary corrections in the manuscript to avoid any confusion.

Regarding the negative sample cycle threshold of less than 50 in Supplementary Figure 3, it should be noted that the negative sample used in this figure refers to *K. pneu* (WT). This WT sample does not contain the sequences targeted by the KPC and IMP primers, but it does contain other DNA sequences common to *K. pneumoniae*. Since this common DNA is not present in the transport media used in Figure 5, a non-specific reaction with the common DNA of *K. pneumoniae* could result in a C_t value below 50.

We have revised the manuscript to include these clarifications and to avoid future misunderstandings.

[Results]

Diagnosis of antibiotic-resistant bacteria using ANCA method

C_{10} represents the cycle threshold of the pure transport media and C_t represents the cycle threshold of the clinical sample.

[Supplementary Information]

Supplementary Fig. 10 (a, b) RT-PCR results of bacteria using (a) KPC- and (b) IMP-specific primers.

(Question 9) The experimental section should be written in more detail. For example, how was the bacteria concentration 10⁵ cells/mL estimated? And how were they spiked in human samples?

Response) Thank you for raising these critical points, and we agree that more detailed information should be provided for clarity.

To prepare the bacterial samples, we first cultured the bacteria according to the established experimental protocol. The optical density (OD) of the bacterial culture was then measured at 600 nm using a nanodrop spectrophotometer. The concentration of bacterial cells was estimated based on a widely accepted relationship in which an optical density (OD₆₀₀) of 1.0 typically corresponds to approximately 8 × 10⁸ cells/mL for many bacterial species.

Once the stock bacterial culture was obtained, it was diluted to the desired concentration. This diluted bacterial suspension was then inoculated into the human samples for subsequent experiments.

We've revised the experimental section of the manuscript to include these details for clarity. We hope that this additional information provides a clearer understanding of our experimental approach.

[Experimental Section]

Direct detection of antibiotic-resistant bacteria using ANCA method. All bacteria clinical isolates were provided by Gyeongsang National University College of Medicine. The bacteria strains were cultured on Luria-Bertani (LB) agar medium at 37 °C for 16 h, then harvested and resuspended in sterilized phosphate-buffered saline (PBS). The optical densities (OD₆₀₀) of the suspensions were determined at 600 nm using a Nanophotometer P330 (Implen, Germany) to facilitate bacterial count estimation.

To verify the applicability of the ANCA method for detecting bacteria in clinical specimens, each bacterial strain was spiked into human urine and blood samples (10⁵ cells/mL). In detail, the 99% bacteria-spiked human sample was prepared by mixing 1 µL of bacteria at a concentration of 10⁷ cells/mL with 99 µL of human sample, and the 90% bacteria-spiked human sample was prepared by mixing 90 µL of human sample with 10 µL of bacteria at a concentration of 10⁶ cells/mL.

(Question 10) What is the limitation of this study? The drawbacks of the study are needed to be discussed.

Response) In the Discussion section, we have taken the opportunity to address some of the limitations of the ANCA method. Although our technique offers several advantages, such as simplicity, rapidity, and the potential for high specificity, it is not without drawbacks. By acknowledging these limitations, we hope to guide future research aimed at refining the ANCA method and expanding its range of applications. We thank the reviewer for encouraging a balanced discussion of this topic.

[Discussion]

The ANCA method represents a pioneering attempt to synergize artificial nucleic acid circuits with Ago proteins. Consequently, its relatively low LOD can be attributed to the early stage of experimental optimization, especially when compared with established modalities for the detection of pathogenic microorganisms. Future avenues of research may explore enhancing the cleavage efficiency of the Ago protein/guide DNA through an iterative accumulation of experimental knowledge. Nevertheless, an evaluation of the C_t values associated with the clinical samples studied in this investigation reveals parallels with C_t values documented in the contemporary clinical literature⁶⁴⁻⁶⁶. Such congruence suggests that the ANCA technique has

adequate robustness for clinical applications. The relatively long reaction time compared to other technologies is a drawback of ANCA. This is inevitable to inhibit the apo-form of the Ago protein. The apo-form is often formed in the absence of guide DNA, causing a non-specific cleavage reaction of dsDNA. To inhibit this, the concentration of Ago protein should not be too high. However, due to the nature of the ANCA technique, the higher the concentration of Ago protein, the more favorable the reaction kinetics. Therefore, it is necessary to choose a concentration condition that moderately addresses both issues, which led us to choose 200 nM Ago protein. However, we are hopeful that this issue will be resolved in the near future. This expectation is based on recent advances in protein engineering, such as inactivated Cas proteins. As research into the function of each domain of the Cas9 nuclease continues, Cas9 nucleases that cut only one strand or inactivated Cas9 nucleases with inhibited cleavage activity have emerged. Given the increased interest in Ago protein research, we anticipate that an Ago protein with an inhibited apo-form will be available. Even in the absence of such protein engineering, the reaction rate can be increased by incorporating additional amplification-type circuitry, such as entropy-driven circuitry or catalytic hairpin assembly. Another drawback is that the data presented in the current NEB guide to DNA design is highly variable. This would explain the difference in sensitivity values between each circuit of the ANCA technology. As more data is collected, it may be possible to detect different target nucleic acids with consistent efficiency.

REVIEWERS' COMMENTS

Reviewer #1 (Remarks to the Author):

This is a significantly revised version of the manuscript. A number of experiments have been performed during revision. Under the optimized conditions, the sensitivity has been improved, increasing approximately 10 times. And, the authors adequately answered the comments from the referees.

Reviewer #2 (Remarks to the Author):

I have read the authors' revised manuscript, and they have addressed my previous questions by including detailed narratives and experimental data. The article now features rich content with practical applicability, meeting my criteria for acceptance.

Reviewer #3 (Remarks to the Author):

In this resubmission, the authors have provided new experimental data to support their statements and carefully revised the manuscript. The authors have satisfactorily addressed my previous concerns.